**Hotplate Precipitation Gauge Calibrations and Field Measurements**
**Nicholas Zelasko [1], Adam Wettlaufer [1], Bujidmaa Borkhuu [1], Matthew Burkhart [1], Leah**
**S. Campbell [2], W. James Steenburgh [2], and Jefferson R. Snider [1, 3]**
**[1] University of Wyoming Department of Atmospheric Science**
**[2] University of Utah Department of Atmospheric Sciences**
**[3] Corresponding Author**

**Abstract –**

First introduced in 2003, approximately 70 Yankee Environmental Systems (YES) hotplate precipitation gauges have been purchased by researchers and operational meteorologists. A version of the YES hotplate is described in Rasmussen et al. (2011; R11). Presented here is indoor- and field-based testing of a newer version of the hotplate; this device is equipped with longwave and shortwave radiation sensors. Hotplate surface temperature, coefficients describing natural and forced convective sensible energy transfer, and radiative properties (longwave emissivity and shortwave reflectance) are reported for two of the new-version YES hotplates. These parameters are applied in a new algorithm, are used to derive liquid-equivalent accumulations (snowfall and rainfall), and these accumulations are compared to values derived by the internal algorithm used in the YES hotplates (hotplate-derived accumulations). In contrast with R11, the new algorithm accounts for radiative terms in a hotplate's energy budget, applies an energy conversion factor which does not differ from a theoretical energy conversion factor, and applies a surface area that is correct for the YES hotplate. Radiative effects are shown to be relatively unimportant for the precipitation events analyzed. In addition, this work documents a 10 % difference between the hotplate-derived and new-algorithm-derived accumulations. This difference seems consistent with R11's application of a hotplate surface area that deviates from the actual surface area of the YES hotplate, and with R11's recommendation for an energy conversion factor that differs from that calculated using thermodynamic theory.

## 1 - Introduction

Two types of instrumentation are available for making point measurements of liquid-equivalent snowfall rates and liquid-equivalent snow accumulations: 1) Weighing gauges and related devices that measure snowfall as it collects in a container or on a surface (Brock and Richardson, 2001; Chapter 9), and 2) optical gauges that measure the concentration and size of snow particles either in free fall or within a wind tunnel (Loffler-Mang and Joss, 2000; Deshler, 1988). Many of these gauges obstruct the wind and thus cause falling snow particles to deflect from the measurement zone. Consequently, rates and accumulations are underestimated and should be adjusted to account for undercatch (Jevons, 1861; Lovblad et al., 1993). Alternatively, both gauge types can be operated within a fenced enclosure that minimizes wind and the resultant undercatch (Goodison et al., 1998; Rasmussen et al., 2012). In addition, optical gauges require a snow particle density to convert concentration and size to a liquid-equivalent rate and accumulation (Brandes et al., 2007; Lempio et al., 2007). Because this density is variable and difficult to measure accurately (Locatelli and Hobbs, 1974), optical snowfall measurements are uncertain and remain uncertain even if undercatch is accounted for. A further disadvantage, for both the weighing and optical devices, is that the entrance to the device can become clogged with snow (Warnick, 1954; Currie, 1998; Stickel et al., 2005).

The Yankee Environmental Systems (YES, 2011) hotplate was developed to minimize the aforementioned uncertainties. Advantages of the hotplate are: 1) it is compact, 2) it is immune to clogging, 3) there is no requirement that snow particles fall through an opening, and 4) the derived rates and accumulations are largely independent of snow particle density, although a dependence does exist (R11; their figure 14). In some applications, a disadvantage of the

hotplate relative to a weighing gauge, is the hotplate's electrical power consumption. This is ~
200 W in Wyoming during winter.
This work furthers efforts to advance the hotplate as a snowfall measurement system
(Borkhuu, 2009; R11; Boudala et al., 2014). We develop calibration constants for two hotplate
systems configured with longwave and shortwave radiation sensors. These are a hotplate gauge
owned by the University of Wyoming (UW) and by the National Center for Atmospheric
Research (NCAR; Boulder, CO) [1]. In addition, we develop a new hotplate data processing
algorithm, derive liquid-equivalent rates and accumulations for 27 precipitation events (snowfall
and rainfall), compare accumulations obtained with the new algorithm to those derived by an
internal algorithm (hotplate-derived accumulations), and compare accumulations to values
derived using weighing gauges.
**2 - Algorithm Development**
The two stacked circular aluminum plates seen in Fig. 1 are the precipitation
measurement portion of the YES hotplate system. The plate diameter ($D_h$) is 0.130 m and both
plates have concentric rings that extend vertically either 3 mm (inner and middle rings) or 1 mm
(outer ring) from the plate surface. One of the plates faces upward and is exposed to
precipitation, the other faces downward. Temperature sensors monitor the top and bottom plates
and feedback-controlled heaters maintain the plates at approximately 75 °C (R11). Electrical
power supplied to the top plate ($Q_{top}$) compensates for power lost via sensible energy, radiative,
and vapor mass transfer. Henceforth, we refer to the latter process as latent power output. The
hotplate-derived wind speed, evaluated using the "factory calibration" discussed in R11, is used

---

[1] When a distinction is needed, we indicate the hotplate, followed by a forward slash, and the location of the deployment. For example, the UW hotplate, deployed at the OWL site, is designated UW/OWL.

in this analysis. The bottom plate power ($Q_{bot}$) is likely a measurement used in the calculation of
that wind speed, but this is speculative because the factory wind speed algorithm is proprietary.
We symbolize this wind speed as $U$ and use it to evaluate a Reynolds number ($Re$), and use the
latter to parameterize sensible energy transfer from the ventilated surface of the top plate. R11
also derived wind speeds by fitting $Q_{bot}$, ambient temperature, and a wind speed measured at 10
m above ground level (AGL). This wind speed is not used in this analysis. The hotplate ambient
temperature ($T$) measurement comes from the sensor seen below the radiation instruments (Fig.
1), the relative humidity ($RH$) measurement comes from a sensor that protrudes below the
electronics box (Fig. 1), and the hotplate pressure sensor is contained within the electronics box.
A complete description of our nomenclature is provided in the Appendix.

Since the hotplate was introduced in 2003, two teams (Borkhuu, 2009; R11) have

reported data processing algorithms. The algorithm in Borkhuu (2009) can be explained by
reference to the equation she used to model the top plate's power budget:

$0 =$                                    Implied Steady-state

$Q_{top}$                              Electrical Power Supplied to Top Plate

$- D_h \cdot K_x \cdot (T_h - T) \cdot (\gamma + \alpha \cdot Re^\beta)$        Sensible Power Output

$- P \cdot E / f_2$                     Latent Power Output                    (1)

In Eq. 1, there are three terms that sum to zero in an assumed steady-state. The last of these, the
latent power output, is proportional to the precipitation rate ($P$) and a snow particle catch
efficiency ($E$) and inversely proportional to $f_2$, an electrical-to-precipitation conversion factor.
Also in Eq. 1, the sensible power output has contributions from natural convection (proportional
to $\gamma$) and forced convection (proportional to $\alpha \cdot Re^\beta$), where $\alpha$, $\beta$, and $\gamma$ are fitted constants. These
convective regimes are discussed in Kobus and Wedekind (1995) and are shown graphically in
their Figure 6. Eq. 1 is similar to the algorithm used by King et al. (1978) to derive cloud liquid
water concentration using a heated airborne sensor.

The algorithm in R11 is based on Eq. 2.

$P = [Q_{top} - Q_{bot} - f_1(U)] \cdot f_2 / E$                        (2)

Here, $f_1(U)$ is a wind speed-dependent function. Also in Eq. 2, we see the conversion factor
introduced in the previous paragraph. Somewhat different from how R11 formulated their
conversion factors for rain and snow, we formulate $f_2$ to account for the warming of ice, melting,
warming of the liquid, and liquid evaporation. For rain, we formulate $f_2$ to account for the
warming of liquid and its evaporation. With an exception that we justify later, we applied the
conversion factors recommended by R11: 1) if $T < 0$ °C, the snow $f_2$ is applied, and 2) if $T > 4$ °C
the rain $f_2$ is applied.

In Eq. 1, the sensible power output is a function of $Re$, and thus $U$, and also a function of

$T$. Hence, Eq. 1 can be rearranged to look similar to Eq. 2 with $P$ dependent on $T$, $U$, $Q_{top}$, $f_2$, and
$E$. A difference between the Eq. 1 and Eq. 2 formulations is the explicit dependence on $Q_{bot}$, in
Eq. 2; this is in addition to the implicit $Q_{bot}$-dependent wind speed in $Re$ (Eq. 1) and in $f_1(U)$ (Eq.

2).

Borkhuu (2009), YES (2011), and R11 surmised that the energetic effect of longwave

and shortwave radiation could, in some settings, be comparable to the latent power output.
Consequently, our hotplate (Wolfe and Snider, 2012) was upgraded to firmware version 3.1.2 in
2011. The upgrade included radiation sensors for the measurement of downwelling longwave
and shortwave fluxes. An objective of this paper is the incorporation of the radiation
measurements into a new precipitation rate algorithm.

We used the following equation to analyze the top plate's power budget:

| 119 | $0 =$ | Implied Steady-state |
| 120 | $Q_{top}$ | Electrical Power Supplied to Top Plate |
| 121 | $- D_h \cdot K_x \cdot (T_h - T) \cdot (\gamma + \alpha \cdot Re^{\beta})$ | Sensible Power Output |
| 122 | $- A_h \cdot \varepsilon_h \cdot \sigma \cdot T_h^{4}$ | Longwave Power Output |
| 123 | $+ A_h \cdot \varepsilon_h \cdot IR_d$ | Longwave Power Input |
| 124 | $+ A_h \cdot (1 - R_h) \cdot SW$ | Shortwave Power Input |
| 125 | $- P \cdot E / f_2$ | Latent Power Output        (3) |

Compared to Eq. 1, Eq. 3 has three additional terms. These describe the interaction of the top
plate with its environment via radiative transfer. Two of these terms are inputs (longwave and
shortwave) and one is an output (longwave).
**2.1 - Hotplate Data Files**

The hotplate outputs data to two files. The previously discussed $Q_{top}$ and $Q_{bot}$ are two

of several recorded variables and both of these are essential for the analysis described here.
One of the files is known as the UHP or "user" hotplate file. The UHP file is provided to all
YES customers. The second file is the SHP or "sensor" file. The SHP file is proprietary but
we were granted access to it by NCAR. Table 1 has the list of all recorded variables and
how some of these are symbolized. A complete list of variables (measured and computed),
and constants, is provided in the Appendix. With the exception of Unix time, all variables in
Table 1 are provided as 60-s averages, sampled at 1 Hz (YES, 2011).


## 2.2 - Radiative Properties

Two radiative properties are applied in our analysis of the top plate's power budget (Eq. 3). In the infrared, or longwave, the emissivity of the top plate is the key property. The material used to fabricate the plates is aluminum, which when exposed to air becomes covered with an aluminum oxide layer. Hence, the hotplate emissivity was taken to be that of oxidized aluminum ($\varepsilon_h = 0.14$; Weast, 1975; Section E). Furthermore, we made two assumptions: 1) the longwave output (Eq. 3) is the product of $\varepsilon_h$, hotplate area ($A_h$), and the flux emitted by a black body at $T_h$, and 2) the longwave input (Eq. 3) is the product $\varepsilon_h$, $A_h$ and the downwelling longwave flux ($IR_d$). In a later section, we explain how we derive $IR_d$.

In the visible, or shortwave, the top plate's reflectance ($R_h$) is the key property. Eq. 3 shows how we factored into the power budget the top plate's reflectance, a measured downwelling shortwave flux ($SW$; Table 1), and $A_h$. A value for $R_h$ was determined as follows. We exposed the UW hotplate to solar illumination, while measuring the solar flux, and then shaded the hotplate to establish a baseline for the determination of $R_h$. During these experiments, there was negligible wind and therefore natural convection dominated forced convection in the budget. The budget equation we used to analyze these measurements has three terms: $Q_{top}$, sensible power output, and solar input. In two experiments, the values $R_h = 0.66$ and $R_h = 0.61$ were derived. We apply the average of these ($R_h = 0.63$) in our analysis of measurements from both UW and NCAR hotplates. Because of the oxide layer, the derived reflectance is smaller than the value reported for polished aluminum reflecting "incandescent" light (0.69; Weast, 1975; Section E) and significantly smaller than the value for vacuum-deposited aluminum at visible wavelengths (0.97; Hass, 1955).

## 3 - Methods

### 3.1 Temperature Measurements

Ice bulb temperatures at OWL were calculated using temperature, $RH$, and pressure measurements made within a fully shielded housing (Steenburgh et al., 2014). At GLE and BTL ice bulb temperatures were calculated using the hotplate-derived temperature, $RH$, and pressure values (Table 1). Because the hotplate temperature sensor is incompletely shielded (Fig. 1), there is concern that its measurement is positively biased by solar heating. We investigated this by differencing hotplate-derived temperatures, acquired during precipitation events at OWL, and values acquired by the fully shielded temperature sensor operated at OWL. On average, the hotplate values were larger ($0.4 \pm 0.4$ $^{\circ}$C). We did not attempt to correct for this bias.

### 3.2 - Site Description

Indoor testing was conducted in a high-bay weather balloon hangar and in a laboratory. These facilities are at the University of Wyoming (UW) and are abbreviated hangar and lab. During wintertime, and especially at night, the hangar is cold ($\sim 0$ $^{\circ}$C); the lab is warm year round ($\sim 20$ $^{\circ}$C). Field measurements (Table 2) were conducted in Southeast Wyoming at the Glacier Lakes Ecosystem Experiments Site (GLE), in Southeast Wyoming near the summit of Battle Pass (BTL), and at the North Redfield site in Western New York (OWL). During both indoor and field measurements, all parameters reported by the hotplate (UHP and SHP variables; section 2.1) were recorded using a custom-built data system.

The accuracy of a hotplate-estimated precipitation rate depends on whether the sensed hydrometeors are rain or snow (R11). We infer the latter using a calculated ice-bulb temperature ($T_{IB}$) (Iribarne and Godson, 1981; Chapter 7). Measurements used to derive the $T_{IB}$s are described in section 3.1. The lower limits on these derived values, assuming the measured $RH$ is

overestimated by 5 % (YES, 2011), is no more than 0.4 ºC colder than the values we report. In
instances with $T_{IB}$s larger than 0 ºC, we assume the sensed hydrometeors were liquid.

### 3.3 - NOAH-II Gauge

The NOAH-II is a weighing-type gauge manufactured by ETI Instrument Systems Inc.
(www.etisensors.com). NCAR operated a NOAH-II at GLE and BTL during 2012, and coauthors
(Campbell and Steenburgh) operated a NOAH-II at OWL (Dec. 2013 through Jan. 2014;
Campbell et al., 2016). The three NOAH-II gauges were outfitted with Alter shields (Goodison
et al., 1998; hereafter G98).

### 3.4 – Indoor Testing

Indoor testing of the UW hotplate was conducted every year from 2011 to 2015; the
NCAR hotplate was only tested in 2012. Based on our testing of the UW hotplate, we have no
evidence indicating that the calibration changed over the duration of any of the field
deployments; however, Wettlaufer (2013) demonstrates that calibration constants did change
over the 2011 to 2015 interval, and likely in response to servicing conducted twice at YES. In
this paper we present calibration constants appropriate for the UW hotplate sensor deployed at
GLE (April 2012) and at OWL (December 2013 through January 2014).
During testing, we controlled the hotplate's radiation environment by placing a material
with known emissivity (painted-steel sheeting, $\varepsilon_s = 0.84$) above and below the hotplate. The steel
sheets were positioned to dominate the hotplate's upward and downward fields of view (Fig. 2);
however, the sheets were positioned so that they were not heated by the hotplate. In that case, the
sheet temperature ($T_s$) can be assumed equal to $T$.


## 3.5 - Downwelling Longwave Flux

As we have already mentioned, previous work concluded that the hotplate method of
determining precipitation can be affected by longwave radiation. In response to that finding,
newer versions of the hotplate have a device that measures longwave radiation (pyrgeometer,
e.g., Albrecht et al., 1974) (Fig. 1 and Table 1).

$$M_{IR} = IR_d - IR_u \qquad (4)$$

The left-hand side of Eq. 4 represents the longwave measurement ($M_{IR}$) and the right-hand side
has the downwelling and upwelling components contributing to $M_{IR}$.

Because $IR_d$ appears in the top plate's power budget (Eq. 3), and since $M_{IR}$ is the only
term in Eq. 4 that is measured, the upwelling component ($IR_u$) must be evaluated. This is
possible because the signal from the pyrgeometer is adjusted, within the hotplate electronics
package, to make the source of the upwelling flux a virtual blackbody at the ambient temperature
(YES 2012, personal communication). In that case, $IR_d$ can be formulated as

$$IR_d = M_{IR} + \sigma \cdot T^4 \qquad (5)$$

where $\sigma$ is the Stefan-Boltzmann constant and $T$ is the hotplate-measured ambient temperature.
We also use Eq. 6 to calculate the downwelling longwave flux

$$IR_d = \varepsilon_s \cdot \sigma \cdot T_s^4 \qquad (6)$$

## 3.6 – Warm-Cold Ambient Temperature Tests

Procedures described here were applied during testing conducted indoors (hangar and lab,
section 3.2) at two different temperatures and are hereafter referred to as the warm/cold test. We
show how values of a warm ($T_w$) and cold ($T_c$) ambient temperature, combined with other

recorded hotplate variables (Table 1), can be used to derive two calibration parameters in Eq. 3
($T_h$ and $\gamma$). In our analysis, the temperature of the steel sheeting ($T_s$) was assumed equal to the
ambient temperature (either $T_w$ or $T_c$) and $IR_d$ was calculated using Eq. 6. By design these tests
had negligible forced-convective and latent energy transfers. In that case, Eq. 7a - b are the top
plate budget equations.
$$0 = Q_{top,w} - D_h \cdot K_x \cdot (T_h - T_w) \cdot \gamma - A_h \cdot \varepsilon_h \cdot \sigma \cdot T_h^4 + A_h \cdot \varepsilon_h \cdot \varepsilon_s \cdot \sigma \cdot T_w^4 + A_h \cdot (1 - R_h) \cdot SW_w \qquad (7a)$$
$$0 = Q_{top,c} - D_h \cdot K_x \cdot (T_h - T_c) \cdot \gamma - A_h \cdot \varepsilon_h \cdot \sigma \cdot T_h^4 + A_h \cdot \varepsilon_h \cdot \varepsilon_s \cdot \sigma \cdot T_c^4 + A_h \cdot (1 - R_h) \cdot SW_c \qquad (7b)$$
The measurements applied in these equations were $T_w$ and $T_c$, the warm and cold plate powers
($Q_{top,w}$ and $Q_{top,c}$), the warm and cold shortwave fluxes ($SW_w$ and $SW_c$), and constants
(Appendix). Values of $T_h$ and $\gamma$ (hereafter referred to as $T_h/\gamma$ pairs) were derived by minimizing
departures from zero simultaneously in Eq. 7a - b. Minimization was conducted using a
Newton's method equation solver (Exelis Visual Information Solutions, Inc.); the convergence
tolerance was $1 \times 10^{-4}$ J s$^{-1}$.
**3.7 - Nusselt-Reynolds Relationship**

The Nusselt number ($Nu = \gamma + \alpha \cdot Re^\beta$), is a component of the sensible power output term

in Eq. 3. In this section, we develop a relationship between $Nu$ and $Re$ based on
measurements recorded in the field when precipitation was not occurring; in a later section
we show how that relationship was applied in the new algorithm.

Conceptually, $Nu$ is a dimensionless representation of the sensible power output. Eq. 8a

was used to calculate $Nu$ with measurements ($Q_{top}$, $T$, and $SW$), a calculated variable ($IR_d$; section
3.5), and constants (Appendix and Table 3).
$$Nu = [Q_{top} - A_h \cdot \varepsilon_h \cdot \sigma \cdot T_h^4 + A_h \cdot \varepsilon_h \cdot IR_d + A_h \cdot (1 - R_h) \cdot SW] / [D_h \cdot K_x \cdot (T_h - T)] \qquad (8a)$$
In the numerator are the terms contributing to the sensible power output, and in the denominator
is a term proportional to the sensible power due to molecular conduction.
Conceptually, $Re$ is a dimensionless representation of the wind speed. Eq. 8b was used to
calculate $Re$ with a measurement ($U$) and constants (Appendix).
$Re = p_x \cdot D_h \cdot U / (R_d \cdot T_x \cdot \mu_x)$ (8b)
Two criteria were used to select a site-specific data subset for the $Nu$-$Re$ development: 1)
no precipitation, and 2) at least three hours of continuous measurements with a broad range of
wind speeds. We fitted the selected $Nu$-$Re$ pairs using a non-linear least squares procedure
(curvefit; Exelis Visual Information Solutions, Inc.); the convergence tolerance for the relative
decrease in chi-squared was $1 \times 10^{-3}$.
**3.8 - Electrical-to-precipitation Conversion Factor**
Equilibrium thermodynamics, with the assumptions that ice melts at $T_o = 0\ ^oC$ and
vaporization occurs at $T_h$, was used to derive the conversion factor in Eq. 3 ($f_2$). Adopting the
temperature criteria from R11 (also see section 2), and a framework from Iribarne and Godson
(1981; Chapter 7), we formulated the theoretical conversion factors as
$f_2(T, T_h) = \{\rho \cdot A_h \cdot [\ C_i \cdot (T_o - T) + L_f(T_o) + C \cdot (T_h - T_o) + L_v(T_h)]\}^{-1}$   ($T < 0\ ^oC$)   (9a)
$f_2(T, T_h) = \{\rho \cdot A_h \cdot [\ C \cdot (T_h - T) + L_v(T_h)]\}^{-1}$    ($T > 4\ ^oC$)   (9b)
This formulation is graphed in Fig. 3a (solid line) where we extended Eq. 9b into the temperature
range ($0\ ^oC < T < 4\ ^oC$) where the distinction between rain and snow is ambiguous (R11).
We now compare the conversion factor derived using Eq. 9a – b with that reported in
R11. To be consistent with R11, we assume $T = T_h = 0\ ^oC$. We find that the ratio of $f_2$ (Eq. 9a)
divided by the factor reported in R11 for snow ($3.99 \times 10^{-8}$ m J$^{-1}$) and the ratio of $f_2$ (Eq. 9b)
divided by the factor reported in R11 for rain ($4.52 \times 10^{-8}$ m J$^{-1}$), are both 0.666. Since these
ratios are equal to the area in R11 ($A_h = 0.008844$ m$^2$), divided by the area applied in our
calculation ($A_h = (\pi/4) \cdot 0.130^2 = 0.01327$ m$^2$), we conclude that the discrepancy is not due to
differing thermodynamic parameters applied in R11's and our calculations (e.g., the latent heat
of vaporization), rather it stems from the different values used for the hotplate area. Further, R11
changed their theoretical $f_2$ to an actual conversion factor that was "..lower because of the
imperfect heat transfer from the precipitation to the hot plate (losses to the air, e.g.)." We do not
find justification for this in R11, nor do we agree with R11's assignment of $A_h = 0.008844$ m$^2$
assuming they were recommending that value for the hotplate sold by YES. Recently, Boudala et
al. (2014) addressed the second of these two points, making it clear that $A_h = 0.01327$ m$^2$ is
appropriate for the hotplate sold by YES.
In light of the above, the ratio of our $f_2$ (Eq. 9a – b with $T = T_h = 0$ °C), divided by the
actual conversion factor in R11, is 0.86 for snow and 0.89 for rain. Since a derived precipitation
rate is proportional to $f_2$ (e.g., Eq. 2), we expect the ratio of a precipitation rate from the new
algorithm (assuming $T = T_h = 0$ °C), divided by a synchronous hotplate-derived precipitation
rate, to be between 0.86 and 0.89. Our expectation hinges on the assumption that the YES
algorithm has incorporated R11's surface area and R11's distinction between theoretical and
actual conversion factors.
We calculate $f_2$ in the new algorithm two ways: 1) In a comparison made to a hotplate-
derived accumulation, our $f_2$ is set to $2.66 \times 10^{-8}$ m J$^{-1}$ (snow) and $3.01 \times 10^{-8}$ m J$^{-1}$ (rain). These
values were obtained from Eq. 9a – b with $T = T_h = 0$ °C and are displayed as a dotted line in Fig.
3a. 2) In comparisons made to either a NOAH-II accumulation or to a laboratory reference
precipitation rate, we evaluate $f_2$ using Eq. 9a – b with a $T_h$ from Table 3 and with the hotplate-
measured ambient $T$ (Table 1). In addition to the step change due to the difference between the
latent heats of sublimation and vaporization, our conversion factor has a weak temperature
dependence (Fig. 3a, solid line). This is due to the warming discussed in section 2. Also, in Fig.
3a we display the actual conversion factor from R11 (dashed line). Our classification of
measurements into snow and rain is discussed in a later section.
**3.9 – Snow Particle Catch Efficiency**

In this section, we evaluate a wind speed-dependent function and use it to account for the

top plate's snow particle catch efficiency ($E$; section 2). The physical processes this function
accounts for are, 1) snow particle bouncing subsequent to collision with the top plate, followed
by transfer away from the top plate by wind, and 2) shearing off of a snow particle after it has
landed on the top plate (R11). This conceptual description of catch on the hotplate is different
from that used to describe catch by weighing gauges where undercatch results because a subset
of snow particles are carried over the gauge by a vertically-accelerated flow (Nespor and Servuk,
1999; Thériault et al., 2012). Both R11 and G98 derive catch efficiencies as the ratio of two
paired values of liquid-equivalent accumulation, one obtained from the gauge of interest and the
other obtained from a second gauge operated inside a Double Fence Intercomparison Reference
Shield (DFIR).

The snow particle catch efficiency functions applied here are both gauge- and location-

specific. For the UW hotplate (at GLE and OWL), and the NCAR hotplate (at BTL), we apply
the function recommended by YES (YES 2012, personal communication; hereafter Y12). Wind
speeds used in the efficiency calculation are the hotplate-derived $U$. In addition, the hotplate
catch efficiency function described by R11 (their Equation 6) was also applied. This was based
on the hotplate $U$ adjusted to the 10-m level with a roughness length $z_o = 0.3$ m (G98, their
Equation 4.3.1) and was only used in analysis of measurements made at OWL. The $z_o$ we picked
corresponds to a surface with "Many trees, hedges, few buildings" (Panofsky and Dutton, 1984;
their Table 6.2). This assignment is consistent with the presence of shrubs and trees (Steenburgh
et al., 2014), and a two-story barn, at the OWL site. The barn was located at the eastern edge of a
fallow field, 80 m west of the gauges at OWL. For the NOAH-II gauge, we applied a function
developed for an 8-inch (diameter) Alter-shielded gauge (G98; their Equation 4.7.1). Wind
speeds used in that calculation are from the hotplate (at GLE and BTL) or from an anemometer
(at OWL) (Campbell et al., 2016). Of course, we are assuming that the function from G98
mimics undercatch by our 12-inch (diameter) Alter-shielded NOAH-II gauge.

In Fig. 3b, we present the three catch efficiency functions (R11 with U adjusted to 10 m,

Y12, and G98). In this graph, the wind speed applied in the R11 function is the value plotted on
the abscissa multiplied by 2.9. This adjustment corresponds to the lowest installation of the
hotplate at OWL and decreases to 2.0 for measurements made after 20131217 [2]. In our
calculation of the R11 catch efficiency functions, the snow depth for the interval of interest
(20131211 to 20140129) was set equal to the average (0.7 m) derived using an ultrasonic snow
depth instrument operated at OWL (Campbell et al., 2016). This average, and the AGL altitudes
of the hotplate installation (Table 2), were used to derive the two wind-speed adjustment factors
(2.9 and 2.0). The basis for this calculation is G98's gauge-height correction formula (their
Equation 4.3.1).

Since the anemometer at OWL was operated at nearly the same height as the top of the

NOAH-II gauge (Steenburgh et al., 2014), and the G98 catch efficiency formula (their Equation

---

[2] AGL altitudes of the two hotplate installations are provided in Table 2.

4.7.1) assumes speeds are measured at the height of the gauge opening, a vertical adjustment of
the wind speed was not factored into the G98 catch efficiencies.

**4 – Testing and Calibration Results**

**4.1 – Warm-Cold Tests**

Results from the warm-cold tests are described here. The derived $T_h/\gamma$ pairs (section 3.6)
are in Table 3. The $T_h$ values are 42.2 °C for the NCAR hotplate deployed at BTL (NCAR/BTL),
52.2 °C the UW hotplate deployed at (UW/GLE), and 65.5 °C for the UW gauge deployed at
OWL (UW/OWL). The first two $T_h$s differ from those presented in Wettlaufer (2013) where, for
the NCAR hotplate, he reported agreement with the nominal plate temperature (75 °C; R11) and
for the UW hotplate (GLE) he reported a larger temperature ($T_h$ = 109 °C). The $T_h/\gamma$ pair
reported in Table 3, for the UW/OWL study, was evaluated after Wettlaufer (2013) reported his
warm-cold test results.
In our analysis of the warm-cold measurements we only used data acquired in the hangar.
As we describe below, this may have improved the accuracy of the resultant $T_h/\gamma$ pairs. This is
because all data needed to derive a $T_h/\gamma$ pair can be obtained without turning off the hotplate.
Wettlaufer (2013) analyzed both hanger and lab data. Both in his work and in ours, the relevant
hotplate properties were derived by averaging over a 5 minute warm interval and a 5 minute cold
interval, and applying these averages in Eq. 7a – b. For us the warm-cold temperature pairings
are 5.4/-4.3 °C (NCAR/BTL), 7.0/-1.1 °C (UW/GLE), and 29.5/10.4 °C (UW/OWL). Compared
to Wettlaufer (2013), our $T_w$s are 15 °C colder (NCAR/BTL and UW/GLE experiments only).
Using our $T_h/\gamma$ pairs (Table 3) and the first two $T_w$s (i.e., for NCAR/BTL and UW/GLE), we
evaluated the term in Eq. 7a representing natural-convective transfer ($D_h \cdot K_x \cdot (T_h - T_w) \cdot \gamma$) and
compared to values derived using $T_h/\gamma$ pairs in Wettlaufer (2013; his Table 2). In the NCAR/BTL
comparison $T_w$ was set at 5.4 °C and in the UW/GLE comparison $T_w$ was set at 7.0 °C. Our
natural-convective term agrees within $\pm$ 0.1 W of those derived by Wettlaufer (2013). Also in
good agreement is the product of $T_h$ and $\gamma$. Relative to Wettlaufer (2013), our $T_h \times \gamma$ product is 6
% larger (NCAR/BTL), and 7 % larger (UW/GLE). We expect that our $T_h/\gamma$ pairs (Table 3),
when applied in Eq. 3, will produce a reasonable estimate of the precipitation rate. We test that
expectation in the next section.

Error limits on $T_h$ and $\gamma$, in Table 3, were derived by perturbing $Q_{top,w}$ (i.e., the value

acquired in the warm test) by $\pm$ 0.5 W and repeating the analysis (Eq. 7a - b). Our estimate of the
$Q_{top,w}$ error ($\pm$ 0.5 W) came from a comparison of values acquired before and after power to the
hotplate was stopped and restarted. These tests were conducted in the hangar and the 10 min
warm up recommended by the manufacturer was adhered to (YES, 2011).
**4.2 - Drip Tests**

This section compares two time sequences of precipitation rate: one calculated with the

new algorithm, the other is the hotplate-derived value (Table 1). The basis for the comparison is
measurements of artificially-produced liquid precipitation made in the hangar.  We applied water
drops to the NCAR and UW hotplates using a volumetric water pump (Ismatec Inc.; Model
7618). Each of these tests has a drip period (4 min) and a nondrip period (5 min). Drops (4 mm
volume-equivalent diameter) were added uniformly to the top plate at a constant volumetric rate.
We convert the pump rate to a reference precipitation rate ($P_{REF}$) and apply the $P_{REF}$ in
subsequent analyses [3]. These drip tests were conducted at $T > 4$ °C.

---

[3] The value of the multiplier that converts the volumetric pump rate ($cm^3$ $min^{-1}$) to precipitation rate (mm $hr^{-1}$) is 4.51.

Because the drip tests were conducted with the hotplate operating as in Fig. 2, and
unventilated, the recorded data were analyzed with $T_s = T$, in Eq. 6 (section 3.5), and with the
sensible power output formulated as $D_h \cdot K_x \cdot (T_h - T) \cdot \gamma$ (Appendix and Table 3). Also, because all
of the pumped water is delivered to the top plate, the catch efficiency is $E = 1$. Hotplate
precipitation rates were derived by inputting measurements ($Q_{top}$, $T$, $U$, and $SW$) and a calculated
variable ($IR_d$; section 3.5) into Eq. 3 and solving for a precipitation rate sequence ($P(t)$). We
symbolize this $P(t)$ as $P_{UW}$ and refer to calculations leading to that sequence as the UW
algorithm. Also, we refer to sequences obtained from the UHP file (Table 1) as $P_{YES}$ and refer to
that calculation as the YES algorithm.
We now compare values of $P_{UW}$ to synchronous values of $P_{YES}$. Typically, these rates
exhibit a maximum ~ 3 min after the nondrip-to-drip transition (Fig. 4). We interpret these
maxima as overestimates, possibly due to a violation of the steady-state assumption. Also
evident, particularly in the $P_{UW}$ sequence, is a minimum. This occurs during the time the
instrument is relaxing to its rest state; i.e. ~ 2 min after a drip-to-nondrip transition. The figure
also demonstrates that thresholding is applied to the $P_{YES}$ sequence, i.e. the YES algorithm
thresholds the output to 0 mm hr$^{-1}$ if values decrease to < 0 mm hr$^{-1}$. This is evident at ~ 16:11
UTC and at three other times in the $P_{YES}$ sequence. In fact, thresholding is not desired for the drip
tests. Thus, the UW sequence is not thresholded in Fig. 4.
Two 1-min averaging intervals are shown in Fig. 4. We set the end of these at the drip-to-
nondrip transitions and symbolize the averages as $<P_{UW}>$ and $<P_{YES}>$. Fig. 5 is a compilation of
the two tests already discussed plus four additional $P_{REF}$ vs $<P_{UW}>$ comparisons and four
additional $P_{REF}$ vs $<P_{YES}>$ comparisons.
We now use linear least-squares regression analysis, and a regression equation of form $y$
$= a \cdot x$, to derive the ratio of two precipitation rates. In Fig. 5 it is apparent that the regression
slope (ratio), derived for the $P_{REF}$ vs $<P_{UW}>$ comparison, does not differ from one by more than $\pm$
1 standard deviation. Ratios for the two hotplates (UW and NCAR) and for three drip tests are
summarized in Table 4. In the third column ($P_{REF}$ vs $<P_{UW}>$), we see that none of the ratios differ
from one by more than $\pm$ 1 standard deviation. Different from Fig. 5 and Table 4, we also
evaluated intercepts of regressions that were not forced through the origin; none of these
intercepts differ significantly from zero (results not shown). From the statistical comparisons in
Table 4, we conclude the $T_h/\gamma$ pairs (Table 3) applied in the UW algorithm (Eq. 3) produce a
precipitation rate consistent with the reference.
Values of the reference rate and the hotplate-derived rate ($<P_{YES}>$) are compared as ratios
in Fig. 5 and in the fourth column of Table 4. These ratios are seen to deviate systematically
from unity, and in the direction discussed in section 3.8. In the unforced regressions (not shown)
the intercepts are negative, but only one of these differed significantly from zero (NCAR/BTL;
intercept = -0.3 $\pm$ 0.1). Negative intercepts are expected because $P_{YES}$ is positively offset, by ~
0.2 mm hr$^{-1}$, during most of the nondrip periods (e.g., 16:21 UTC in Fig. 4).
**5 - Field Measurements**
This section is organized as follows: Section 5.1 presents field measurements of ambient
temperature and ambient ice-bulb temperature. We use this information to classify 27
precipitation events as snowfall or rainfall. Section 5.2 presents the $Nu$-$Re$ relationship we use to
account for the sensible power output in Eq. 3. Section 5.3 describes how we derive a
precipitation rate for a hotplate based on measurements made in the field. Section 5.4 compares
time-integrated precipitation rates (accumulations) derived using the two algorithms. In section
5.4, we also compare hotplate accumulations to values from the NOAH-II.


## 5.1 – Field-measured Temperatures and Ice-bulb Temperatures


The 27 precipitation events are summarized in Table 5. Measurements were made during

2012, at the two Southeast Wyoming field sites (BTL and GLE), and during 2013 and 2014 at
the Western New York site (OWL). Table 5 and Fig. 6 have event-averaged ambient
temperatures ($<T>$) and event-averaged ambient ice-bulb temperatures ($<T_{IB}>$). Twenty-three of
the events have $<T> \leq$ -3.3 $^{\circ}$C and upper-limit temperature ($<T>$ plus two standard deviations) no
warmer than -2.3 $^{\circ}$C. We classified these as snowfall. In addition, we classified four events as
rainfall. These had $<T_{IB}> \geq$ +2.9 $^{\circ}$C and lower-limit temperature ($<T>$ minus two standard
deviations) no colder than +2 $^{\circ}$C.

## 5.2 - Nusselt-Reynolds Relationship


Fig. 7b shows a plot of the *Nu-Re* fit function with the data used to constrain the function.

This result is based on UW hotplate measurements (GLE site) and formulas developed in
section 3.7. Fit coefficients ($\alpha$, $\beta$, and $\gamma$) are reported in Table 6 for each field site. Hansen and
Webb (1992) reported $\alpha = 0.09$ and a $\beta$ between 0.69 and 0.72 for a surface similar to the
hotplate (circular with three concentric rings); however, their flow direction was perpendicular to
the plate surface. The values of $\alpha$ and $\beta$ we report may differ from those in Hansen and Webb
(1992) because the flow is principally parallel to the plate surface at our field sites. There are two
other differences relative to Hansen and Webb (1992): 1) Our geometrically-averaged *Nu* (~ 360)
is about a factor of five larger, and 2) our *Re* extends over a much larger range. Finally, we note
that compared to Fig. 7b there is an order of magnitude narrower *Re* range in the NCAR/BTL
and UW/OWL *Nu-Re* plots (not shown).
Fig. 7a is a companion to Fig. 7b showing the $\gamma$ based on the warm-cold test. The error
limit on this datum is explained in section 4.1. Since $Nu$ is dependent on the $T_h$ derived in the
warm-cold test (section 3.6), we expect the $Nu$-$Re$ function to converge to the warm-cold $\gamma$ in the
limit of small $Re$. In our assessment of convergence, we evaluated the limiting $Nu$ at the $Re$
corresponding to the minimum $U$ reported in the hotplate data output (0.1 m s$^{-1}$). This minimum
$U$ establishes the left end of the function in Fig. 7b. Convergence of the $Nu$-$Re$ relationship to
within the error limit on the warm-cold $\gamma$, at the former's left-most limit, is evident in Fig. 7a – b.
Convergence is also evident in the NCAR/BTL and UW/OWL plots analogous to Fig. 7a – b
(not shown) and this in spite of narrower $Re$ range in those datasets.
**5.3 - Precipitation Rate from Field Measurements**
Fig. 8 shows budget terms (Eq. 3) for one of the four rainfall events in our dataset (OWL-
15). The three output terms (sensible, latent, and longwave), and three input terms (top plate,
longwave, and shortwave) are shown in Fig. 8a - b. In this section we begin with the latent power
output (i.e., $P \cdot E / f_2$ in Fig. 8a) and describe how we calculate the rainfall rate. We also contrast
that calculation with steps followed in the case of snowfall.
The first step in the calculation is conversion of the latent power output term (Fig. 8a) to
a provisional precipitation rate; this is done by multiplying each element of the term by the
corresponding element of $f_2$ (Eq. 9b). This operation is referred to as element-by-element vector
multiplication. Thresholding is applied next. Both a 300-s running average of the provisional rate
and a 10-s running average of the provisional rate are computed. If the 300-s average exceeds
0.25 mm hr$^{-1}$, and the 10-s average exceeds 0 mm hr$^{-1}$, the rate is stored as the 10-s average;
otherwise the rate is stored as 0 mm hr$^{-1}$. We refer to the resultant as $P_{UW}$, but we note that in
section 4.2 the $P_{UW}$ sequences are unthresholded. Both the thresholded and unthresholded
sequences are presented in Fig. 8c – d. The thresholded $P_{UW}$ is identical to the unthresholded $P_{UW}$
where the 300-s average exceeds 0.25 mm hr$^{-1}$ and the 10-s average exceeds 0 mm hr$^{-1}$.

In the case of snowfall, the $f_2$ is calculated using Eq. 9a and applied as discussed in the

previous paragraph. Finally, the precipitation rate is derived as the resultant of element-by-
element vector multiplication of the thresholded $P_{UW}$ and the reciprocal of the snow particle
catch efficiency (section 3.9).
**5.4 – Comparisons of Liquid-equivalent Accumulation**

Here we use linear least-squares regression analysis, with a regression equation of form $y$

$= a·x$, to derive the ratio of two measures of liquid-equivalent accumulation for snow. In Fig. 9,
these measures are the accumulations derived using the UW and YES algorithms. In the these
algorithms the particle catch efficiency function is the one described in Y12 and $f_2$ is $2.66 \times 10^{-8}$
m J$^{-1}$ (section 3.8). The data points correspond to measurements made at GLE (UW hotplate), at
BTL (NCAR hotplate), and at OWL (UW hotplate). We note that 19 of 23 $y$-axis values are from
the same instrument (UW hotplate) and are derived using the same calibration (UW/OWL) used
to produce the result shown in the third row of Table 4. Statistical consistency between the ratio
in Fig. 9 ($0.79 \pm 0.05$) and the ratio in the third row of Table 4 (i.e., $0.79 \pm 0.03$ for the $P_{REF}$ vs
$<P_{YES}>$ ratio) suggest a systematic error in the YES-derived precipitation rates and
accumulations. This assertion is reinforced by the three NCAR hotplate points straddling the
best-fit line, in Fig. 9, and by the ratio reported in Table 4 for the NCAR hotplate (i.e., $0.81 \pm$
0.03 for the $P_{REF}$ on $<P_{YES}>$ ratio). However, we cannot exclude the possibility that bias in our
field-based calibration coefficients ($\alpha$, $\beta$, and $\gamma$; Table 6) is the reason for a UW/YES ratio
significantly smaller than unity in Fig. 9.
As was discussed in section 4.2, and demonstrated in Fig. 4, during the indoor nondrip
periods the $P_{YES}$ sequence is positively offset. A plausible reason for this, and for the ratios < 1
reported in the previous paragraph, is disregard for longwave forcing in the YES algorithm.
Since we do not have access to the YES algorithm, we estimated the longwave radiative effect
by setting the longwave terms to zero in Eq. 3. After doing this, a larger UW/YES ratio (a = 0.83
± 0.04) was obtained in a plot analogous to Fig. 9. From this modest increase of the UW/YES
ratio, we conclude that longwave forcing *cannot* explain the shift of the best-fit line away from
unity in Fig. 9. An even smaller perturbation of the UW/YES ratio was obtained in calculations
that set the shortwave term to zero in Eq. 3 (results not shown).
Further evidence for systematic error in the YES values comes from Fig. 10. With the
exception that these data are for rain observed at OWL (section 5.1) the comparison in Fig. 10 is
similar to Fig. 9. Although the number of points is small, Fig. 10 establishes that our finding of a
UW/YES ratio significantly smaller than unity is true for both rainfall and snowfall. In addition,
Fig. 11 strengthens this conclusion by showing agreement between values of the UW
accumulation and the NOAH-II accumulation when both gauges detected rain.
An additional assessment of snowfall at OWL is presented in Fig. 12a – c. In these graphs
the NOAH-II measurements are plotted on the abscissa and different interpretations of the UW
hotplate measurements are plotted on the ordinate. For both devices, we plot the ratio of a liquid-
equivalent accumulation divided by an event-averaged catch efficiency, and we note that the
numerator of these ratios are accumulations that were not corrected for inefficient catch [4]. Table
5 demonstrates two features of the OWL snow data set: 1) The event-averaged catch efficiency
based on Y12 (*<E Y12>*) is consistently larger than the event-averaged efficiency based on R11

---

[4] This comparison was also made using accumulations corrected with a time-dependent catch efficiency (section 5.3), but we found that the fit-line slopes differed by less than ± 5 % from those in Fig. 12.

($<E\ R11>$), and 2) the event-averaged efficiency $<E\ R11>$ is comparable to $<E\ Y12\ An>$, where
the latter is the event-averaged efficiency derived with the anemometer $U$ and the Y12 catch
efficiency function. These features are consistent with the altitude adjustment in R11, which
increases the wind speed (section 3.9), and thus decreases $<E\ R11>$ relative to $<E\ Y12>$. They
are also consistent with a low bias in the hotplate-derived $U$. The latter is supported by a
comparison of the hotplate $U$ vs anemometer $U$ where the fit-line slope is $0.55 \pm 0.05$ for the 19
snow events at OWL (results not shown).

Consistent with the ranking of event-averaged values of $E$ (Table 5), Fig. 12a shows that

the hotplate values, derived with the hotplate $U$ and the Y12 catch efficiency function, are
smaller (on average) than the NOAH-II-derived values. We also see that the 15% underestimate
in the hotplate (Fig. 12a) reverses to a slight overestimate when using the R11 catch efficiency
function (Fig. 12b) and when using the anemometer $U$ with the Y12 function (Fig. 12c). These
results do not allow us to specify contributions to the 15% underestimate (Fig. 12a), coming
from the fact that the Y12 function does not use a height-adjusted $U$, or from the suspected
hotplate underestimate of $U$. Further studies focused on development of a hotplate catch
efficiency function dependent on the local wind speed, as opposed to the wind speed at 10 m
(R11), and investigation of the hotplate's determination of wind speed, are needed to resolve this
issue. Since there is error in the NOAH-II values used in this comparison, there is also need for
characterization of that uncertainty (random and systematic). Error can propagate from the
NOAH-II measurements themselves and from the catch efficiency function we applied to those
data (section 3.9).


## 6 - Conclusions

Starting with measurements acquired from two YES hotplates, we derived precipitation

rates and accumulations for 27 snowfall and rainfall events. The basis for this is a power budget

equation similar to that in King et al. (1978). We changed the budget equation by including terms

describing the longwave and shortwave radiant energy transfers. To the best of our knowledge,

this is the first time that radiative terms have been incorporated into a hotplate data analysis

algorithm and reported in the scientific literature.

We demonstrated that radiative forcing of the budget is relatively unimportant for the

precipitation events analyzed. This is because the top plate's shortwave absorptance (i.e., $1 - R_h$

in Eq. 3), and its longwave emissivity, are small compared to unity, because a majority of events

occurred at night, and because generally overcast conditions diminished the significance of the

longwave forcing.

In this paper, we used computational methods different from those in R11, and we

derived and applied different calibration coefficients. In spite of these changes we report

precipitation rates and accumulations that strongly correlate with the output of two YES

hotplates. However, a systematic difference is evident in our comparisons of the UW and YES

algorithms. We surmise that the difference comes from the following: 1) R11's assignment of $A_h$

(0.00884 m$^2$ vs 0.01327 m$^2$ in the UW algorithm), 2) R11's distinction between a theoretical and

an actual energy conversion factor, and 3) the incorporation of #1 and #2 into the YES algorithm.

Clearly, R11's $A_h$ is not justified for hotplates sold by YES (Boudala et al., 2014; YES 2017,

personal communication). R11's distinction between conversion factors is more problematic.

That distinction can be interpreted two ways: either 1) The distinction accounts for

environmental thermal energy input that assists the conversion of precipitation mass to vapor, or
2) the distinction accounts for the loss of snow particles from the top surface of the hotplate due
to removal by wind. Because early in the warming process a precipitation element attains a
temperature larger than that of the air, we assert that the first of these phenomena is unlikely to
contribute significantly to the energy budget. The second may be significant, but it is our opinion
that removal of precipitation mass by wind is best accounted with a catch efficiency, not with a
distinction between conversion factors. Lastly, accounting for either of these phenomena,
independent of an adjustment of the catch efficiency, should be accomplished with an increase of
an actual conversion factor relative to the theoretical value, not with the decrease proposed by
R11.
**Acknowledgements -**
We gratefully acknowledge the support provided by the UW Department of Atmospheric
Science Engineering Group. We also thank Ontario Winter Lake-effect Systems (OWLeS)
project PIs Bart Geerts and Dave Kristovich for their leadership and Philip Bergmaier for
maintaining the UW hotplate during OWLeS. This work was supported by the United States
National Science Foundation (Award EPS 1208909), the Wyoming Water Development
Commission (Award WWDC40395H), and by the U.S. Department of Interior (Award
1000628L).

**Appendix – Nomenclature**
$A_h$        Area of YES hotplate = 0.01327 m$^2$
$C$        Liquid H$_2$O specific heat capacity = 4218 J kg$^{-1}$ K$^{-1}$ (assumed independent of

temperature; Iribarne and Godson, 1981; their Table IV-5)

$C_i$        Solid H$_2$O specific heat capacity = 2106 J kg$^{-1}$ K$^{-1}$ (assumed independent of

temperature; Iribarne and Godson, 1981; their Table IV-5)

$D_h$        Diameter of YES hotplate = 0.130 m
$E$        Snow particle catch efficiency (section 3.9)
$f_1$        Wind speed-dependent property in Eq. 2 [W]
$f_2$        Electrical-to-precipitation conversion factor [m J$^{-1}$]
$IR$        Upwelling or downwelling component of longwave flux [W m$^{-2}$]
$L_f(T_o)$        Latent heat of fusion evaluated at the thermodynamic reference temperature =

0.3337x10$^6$ J kg$^{-1}$ (Iribarne and Godson, 1981; their Table IV-5)

$L_v(T_h)$        Latent heat of vaporization at $T_h$ (Iribarne and Godson, 1981; their Equation

4.103) [J kg$^{-1}$]

$M_{IR}$        Measured longwave flux (section 3.5) [W m$^{-2}$]
$Nu$        Nusselt number
$P$        Liquid-equivalent precipitation rate [mm hr$^{-1}$ or m$^3$ m$^{-2}$ s$^{-1}$]
$P_{Ref}$        Reference precipitation rate (section 4.2) [mm hr$^{-1}$ or m$^3$ m$^{-2}$ s$^{-1}$]
$P_{UW}$        Precipitation rate derived with UW algorithm (section 5.3) [mm hr$^{-1}$ or m$^3$ m$^{-2}$ s$^{-1}$]
$P_{YES}$        Precipitation rate derived with YES algorithm (section 4.2) [mm hr$^{-1}$ or m$^3$ m$^{-2}$ s$^{-1}$]
$Q_{bot}$        Bottom plate power [W]
$Q_{top}$        Top plate power [W]
$R_d$        Dry air specific gas constant = 287 J kg$^{-1}$ K$^{-1}$
$Re$        Reynolds number
$R_h$        Hotplate Reflectance = 0.63 (section 2.2)
$SW$        Measured shortwave flux (section 2.2) [W m$^{-2}$]
$T$        Ambient temperature [$^o$C or K]
$T_h$        Hotplate surface temperature (section 3.6) [$^o$C or K]
$T_o$        Thermodynamic reference temperature = 0.0 $^o$C
$T_s$        Temperature of painted-steel sheeting [$^o$C or K]
$U$          Wind speed [m s$^{-1}$]
**Greek Symbols**
$\alpha$          Fitted *Nu-Re* Coefficient (section 3.7)
$\beta$          Fitted *Nu-Re* Coefficient (section 3.7)
$\varepsilon_h$          Hotplate emissivity = 0.14 (section 2.2)
$\varepsilon_s$          Emissivity of painted-steel sheeting = 0.84 (section 3.4)
$\gamma$          Coefficient derived in warm-cold tests (section 3.6) or a coefficient in the *Nu-Re*

relationship (section 5.2)

$\rho$          Liquid $H_2O$ density = 1000 kg m$^{-3}$ (assumed independent of temperature)
$\sigma$          Stefan-Boltzmann constant = 5.67x10$^{-8}$ W m$^{-2}$ K$^{-4}$
**Subscripts**
*c*          Indoor cold setting
*d*          Downwelling
*h*          Hotplate
*IB*          Ice-bulb
*s*          Painted-steel sheeting
*u*          Upwelling
*w*          Indoor warm setting
*x*          Property of air film adjacent to the hotplate surface: $p_x$ = standard-atmosphere

pressure at the altitude of the measurement. The following three film properties

are held constant in calculation of the Reynolds number (section 3.7) and in

calculation of the sensible power output due to molecular conduction (section

3.7): 1) temperature ($T_x$ = 303.15 K), 2) dynamic viscosity ($\mu_x$ = 1.862x10$^{-5}$ kg m$^{-}$

$^1$ s$^{-1}$; Rogers and Yau (1989; their Table 7.1)), and 3) thermal conductivity ($K_x$ =

2.63x10$^{-2}$ J m$^{-1}$ s$^{-1}$ K$^{-1;}$ Rogers and Yau (1989; their Table 7.1)).



**Operator**

$<y>$        Time average of property $y$

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

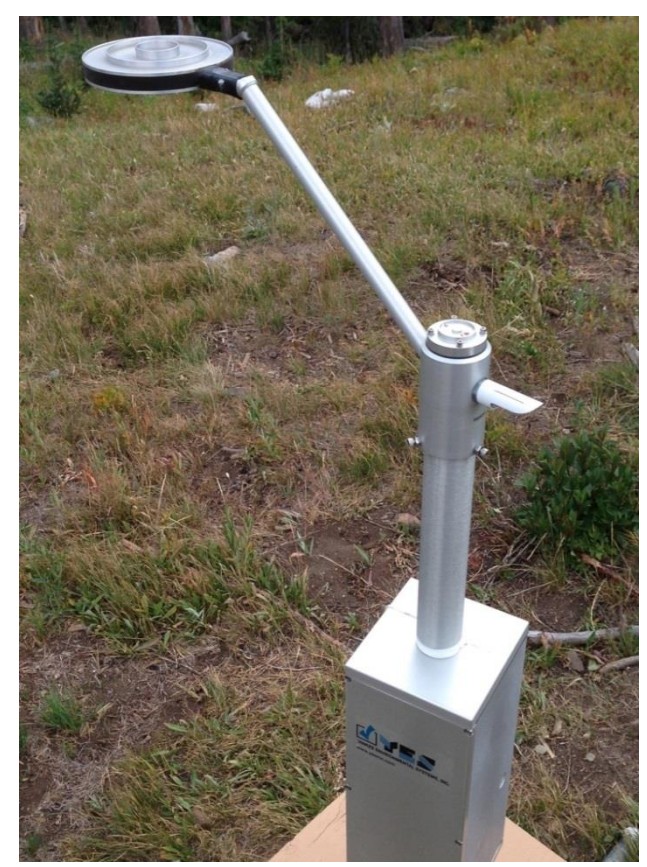

Top and Bottom Plates
(Precipitation Sensor)

Longwave and Shortwave
Radiation Sensors

Temperature Sensor

Electronics

Figure 1 – The Yankee Environmental Systems TPS-3100 Total Precipitation Sensor

with longwave and shortwave radiation sensors.













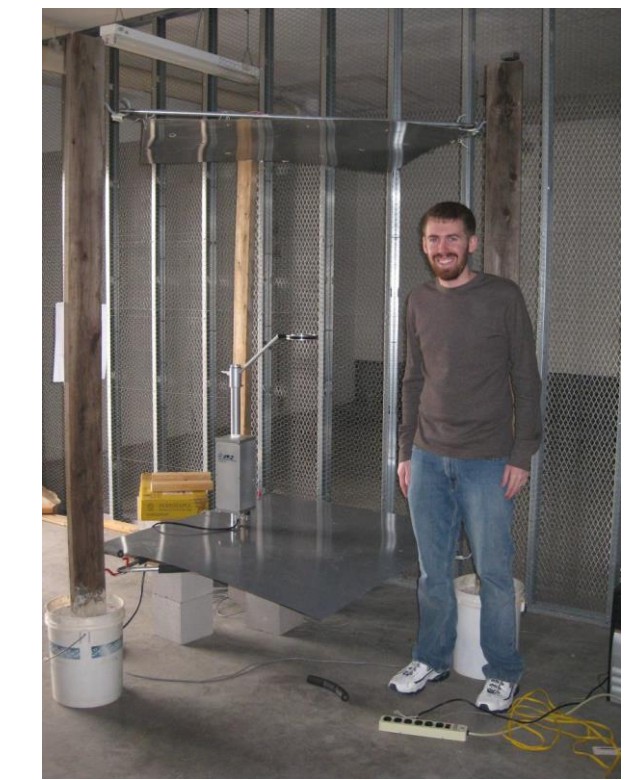

Figure 2 – Picture taken during indoor testing showing a hotplate's precipitation sensor
positioned between top and bottom painted-steel sheets.

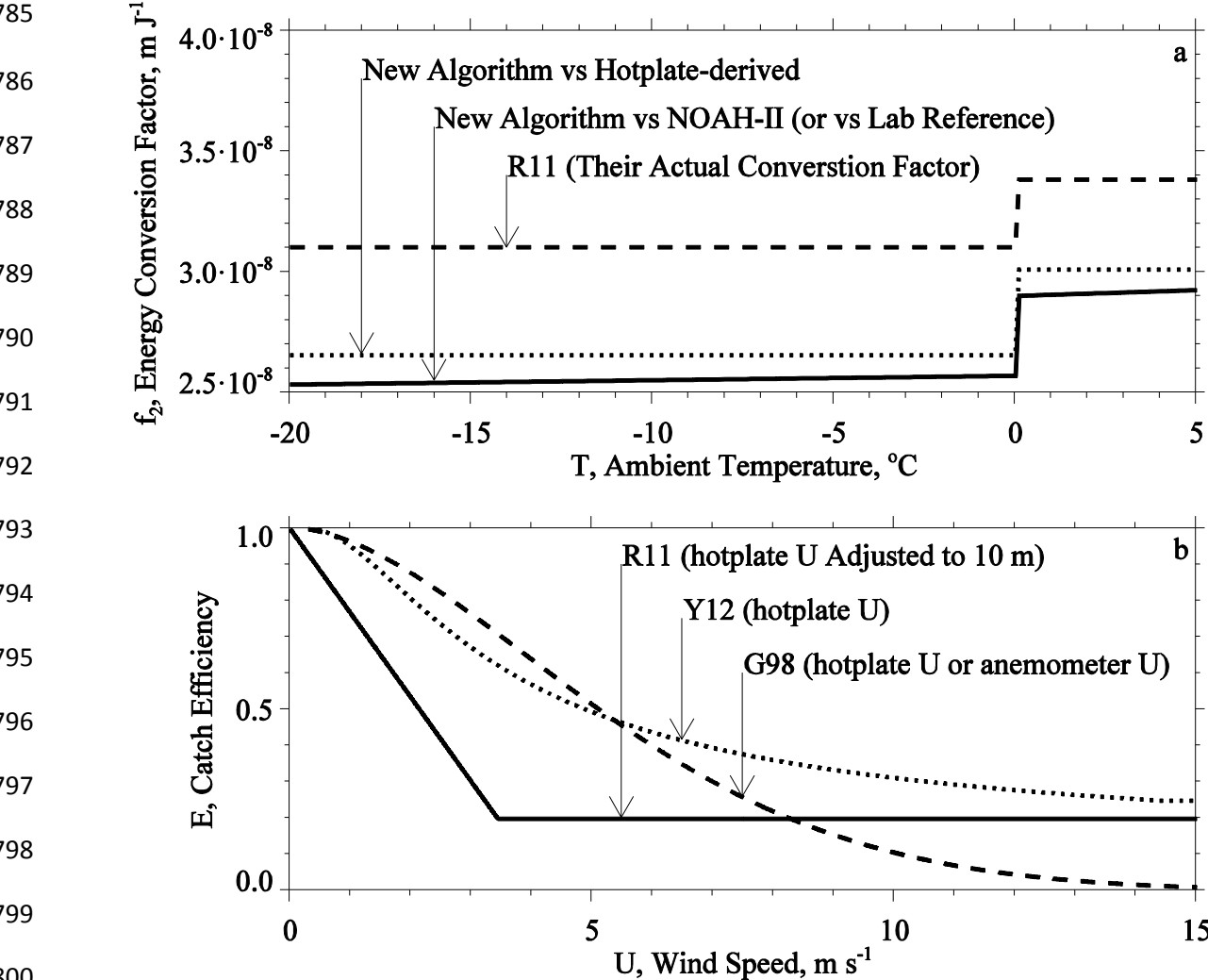

Figure 3 – a) Electrical-to-precipitation conversion factors vs ambient temperature assuming snow at $T < 0$ °C and rain at $T > 0$ °C. See text for details. b) Snow particle catch efficiency vs wind speed using the R11, Y12, and G98 formulations discussed in the text.

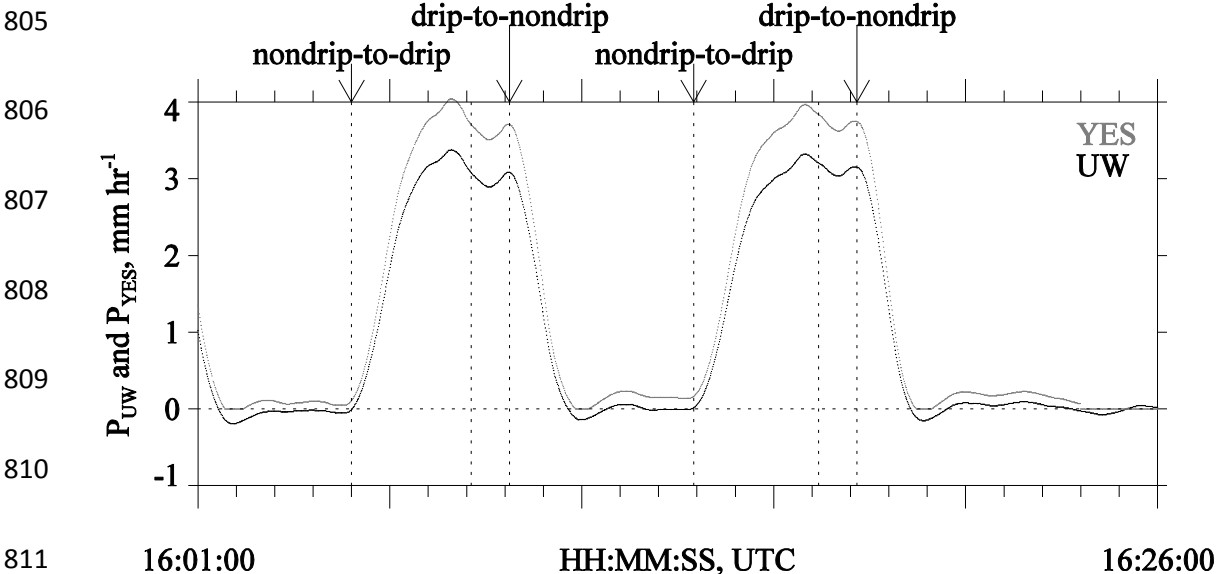

16:01:00                           HH:MM:SS, UTC                     16:26:00

Figure 4 – Precipitation rates, derived using the UW and YES algorithms, plotted vs time.

Dashed vertical lines illustrate nondrip-to-drip transitions, drip-to-nondrip transitions, and one-
minute precipitation averaging intervals. In this figure, the one-minute averaging intervals are ~
16:08 to ~ 16:09 UTC and ~ 16:17 to ~ 16:18 UTC. Measurements are from the UW hotplate
operating indoors on 20120229. The UW/GLE calibration constants (Table 3) and an $f_2$ derived
with the second of two methods (section 3.8) were applied in the UW algorithm.

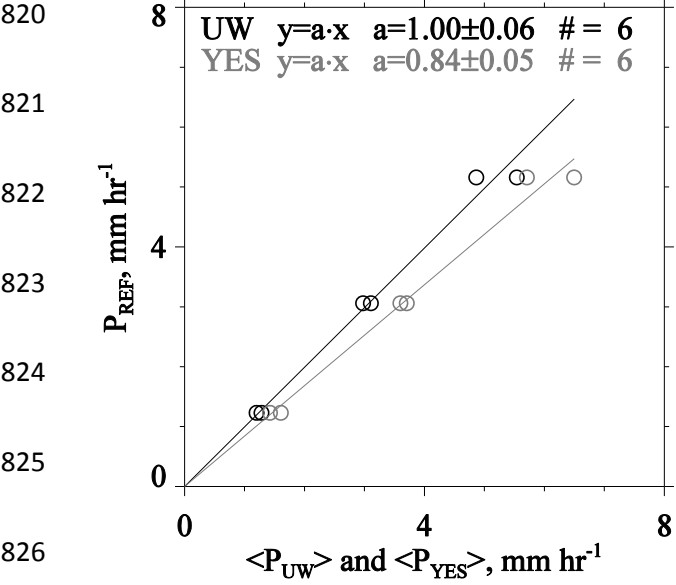

Figure 5 - Reference precipitation rate vs time-averaged $P_{UW}$ and $P_{YES}$. Measurements are from the UW hotplate operating indoors on 20120229. The UW/GLE calibration constants (Table 3) and an $f_2$ derived with the second of two methods (section 3.8) were applied in the UW algorithm. Regression lines were forced through the origin and $x$ deviations (horizontal departures of data from regression line) were used as the basis for the least squares criterion of best fit (Young, 1962). Standard deviations on the fitted ratios (confidence intervals) were derived using Student's t-distribution at the 95% level (Havilcek and Crain, 1988).

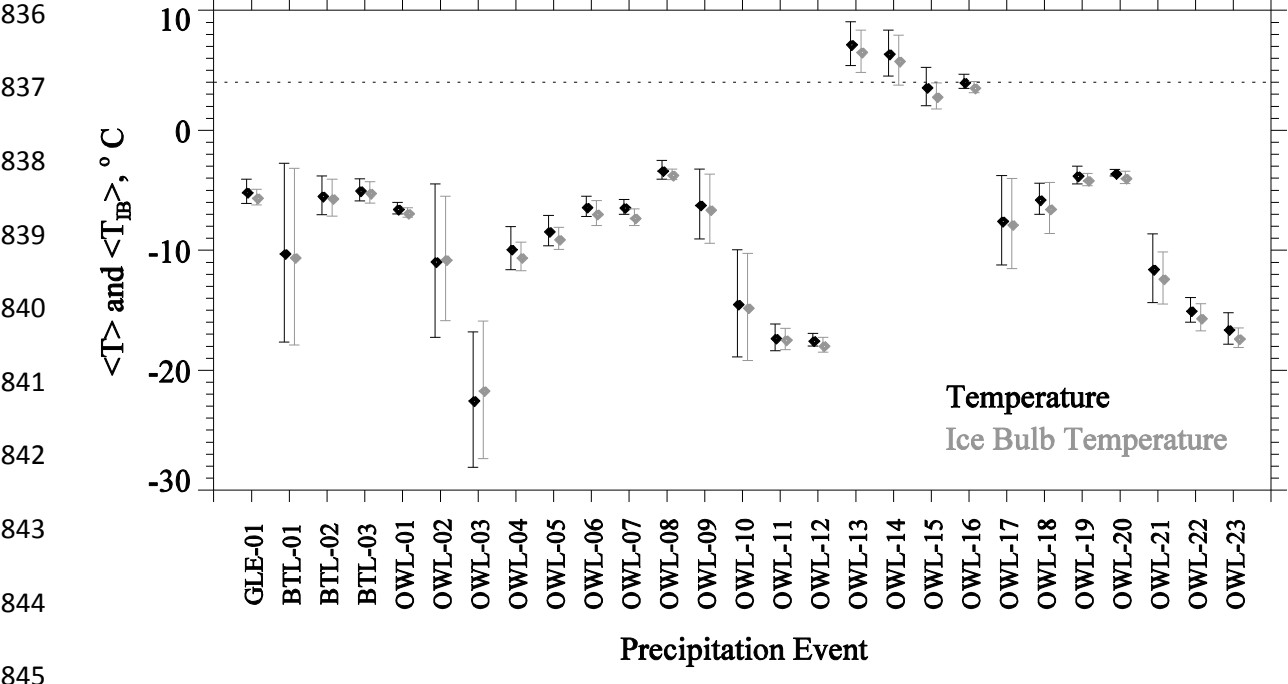

Figure 6 – Event-averaged ambient temperature ($<T>$) and event-averaged ambient ice bulb temperature ($<T_{IB}>$). The abscissa shows the 27 precipitation events in the order presented in Table 5. Error bars are ± 2 standard deviations. The dashed horizontal line is drawn at +4 °C.

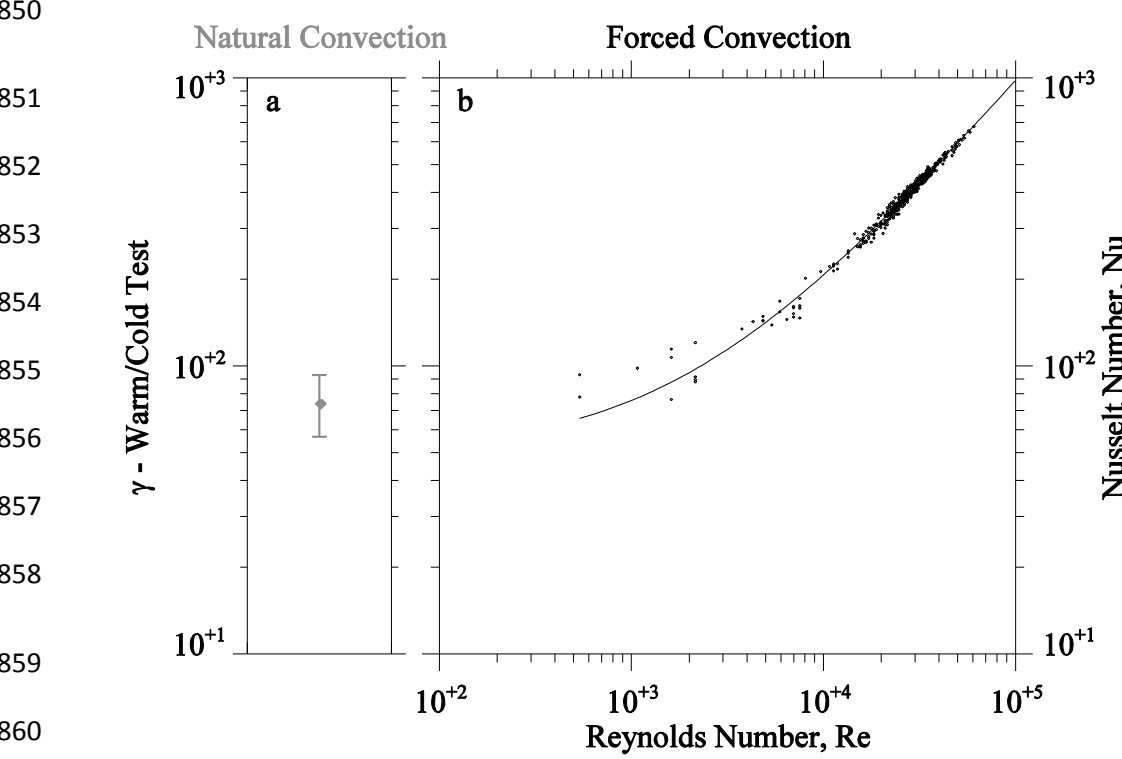


Figure 7 – a) γ from the warm-cold test summarized in the second row of Table 3. Error
limits were derived by perturbing $Q_{top,w}$ (i.e., the value acquired in the warm test) by ± 0.5 W and
repeating the analysis based on Eq. 7a - b.  b) *Nu* vs *Re* scatterplot and fit curve for the UW
hotplate at the GLE site. For clarity, only every fortieth *Nu-Re* data pair is plotted. The minimum
*Re* plotted (data and fit function) corresponds to the minimum *U* reported in the UHP file (0.1 m
s$^{-1}$ ). The measurement interval is 20120402 04:00 UTC to 20120402 09:00 UTC at the GLE site.
The UW/GLE $T_h$ (Table 3) was applied in the data analysis.

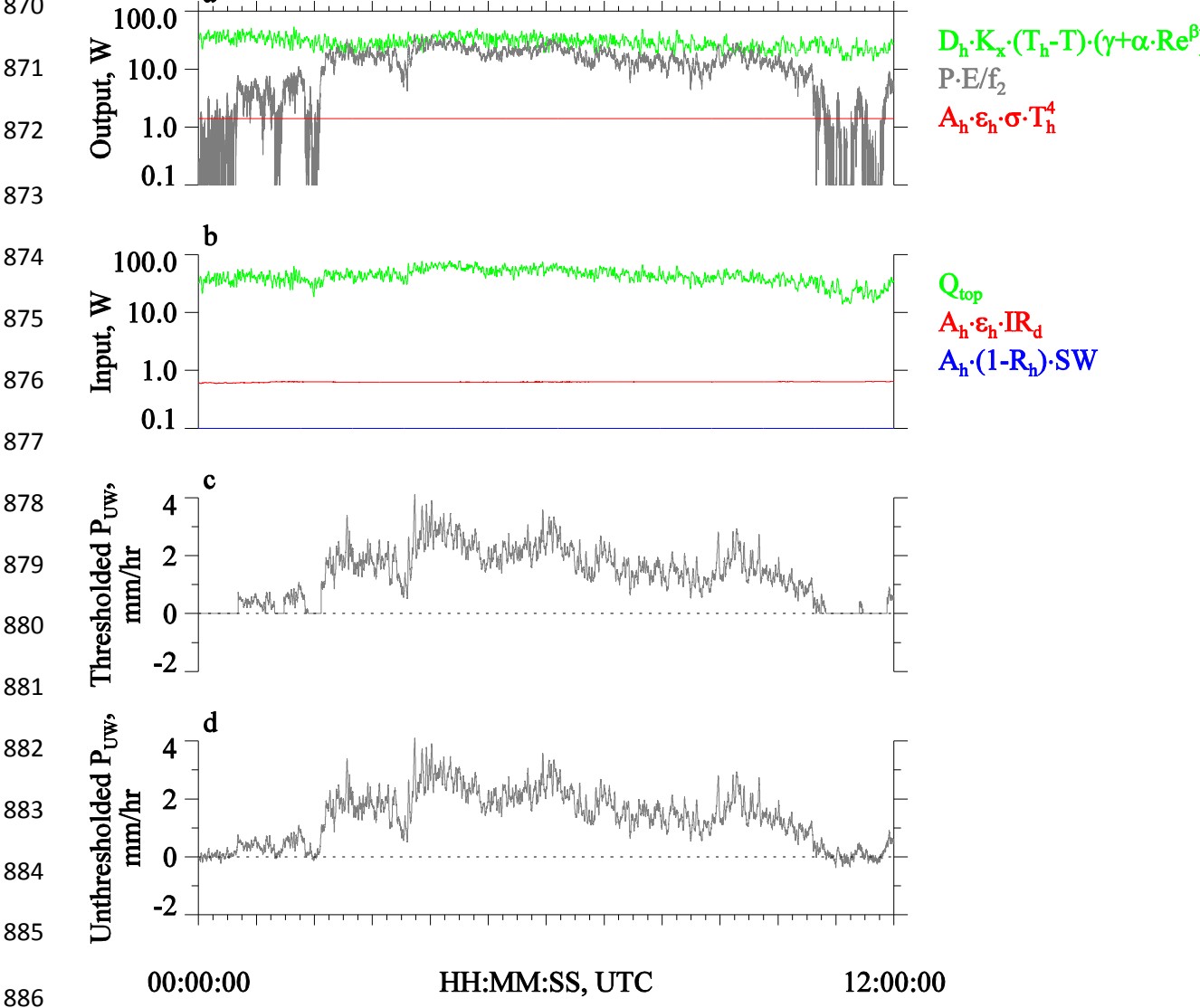

Figure 8 – Hotplate properties during rain (event = OWL-15). Because this event

classifies as rain, $E = 1$ was applied in the UW algorithm. a) Budget output terms (Eq. 3); i.e., the

sensible, latent, and longwave outputs. b) Budget input terms (Eq. 3); i.e., top plate, longwave,

and shortwave inputs. The shortwave term is zero for this nighttime example, but is set to 0.1 W

in the plot. c) Thresholded precipitation rate. d) Unthresholded precipitation rate.

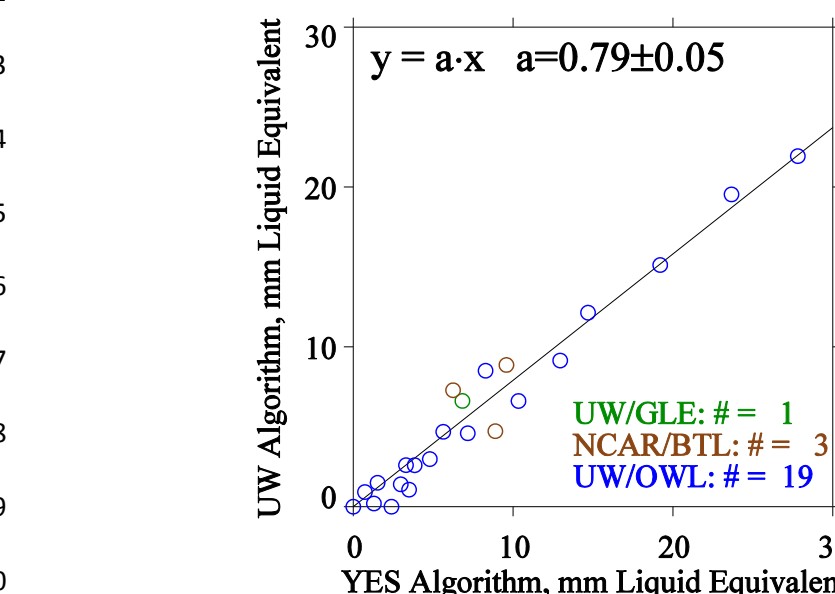

Figure 9 – Snow accumulations derived using the UW algorithm vs snow accumulations derived using the YES algorithm. Both the Y12 catch efficiency function and an $f_2$ derived with the first of two methods discussed in section 3.8 were applied in the UW algorithm. The regression line was forced through the origin and $y$ deviations (vertical departures of data from regression line) were used as the basis for the least squares criterion of best fit (Young, 1962). The standard deviation on the fitted ratio (confidence interval) was derived using Student's t-distribution at the 95% level (Havilcek and Crain, 1988).

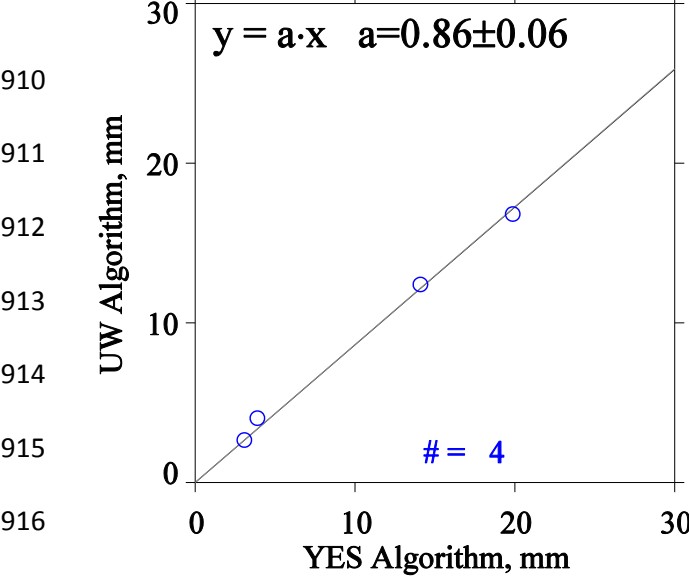

Figure 10 – Rain accumulations derived using the UW algorithm vs rain accumulations

derived using the YES algorithm. An $f_2$ derived with the first of two methods discussed in section

3.8 was applied in the UW algorithm. The regression line was forced through the origin and $y$

deviations (vertical departures of data from regression line) were used as the basis for the least

squares criterion of best fit (Young, 1962). The standard deviation on the fitted ratio (confidence

intervals) was derived using Student's t-distribution at the 95% level (Havilcek and Crain, 1988).

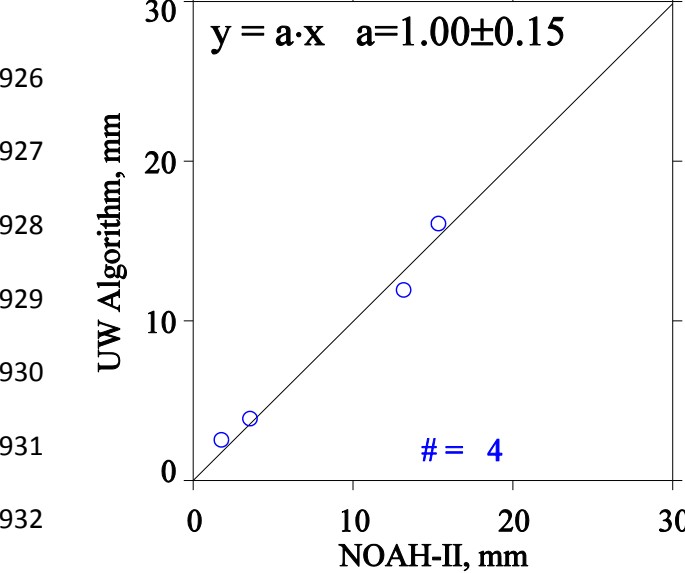

Figure 11 –Rain accumulations derived using the UW algorithm vs rain accumulations from the NOAH-II gauge. An $f_2$ derived with the second of two methods discussed in section 3.8 was applied in the UW algorithm. The regression line was forced through the origin and y deviations (vertical departures of data from regression line) were used as the basis for the least squares criterion of best fit (Young, 1962). The standard deviation on the fitted ratio (confidence intervals) was derived using Student's t-distribution at the 95% level (Havilcek and Crain, 1988).

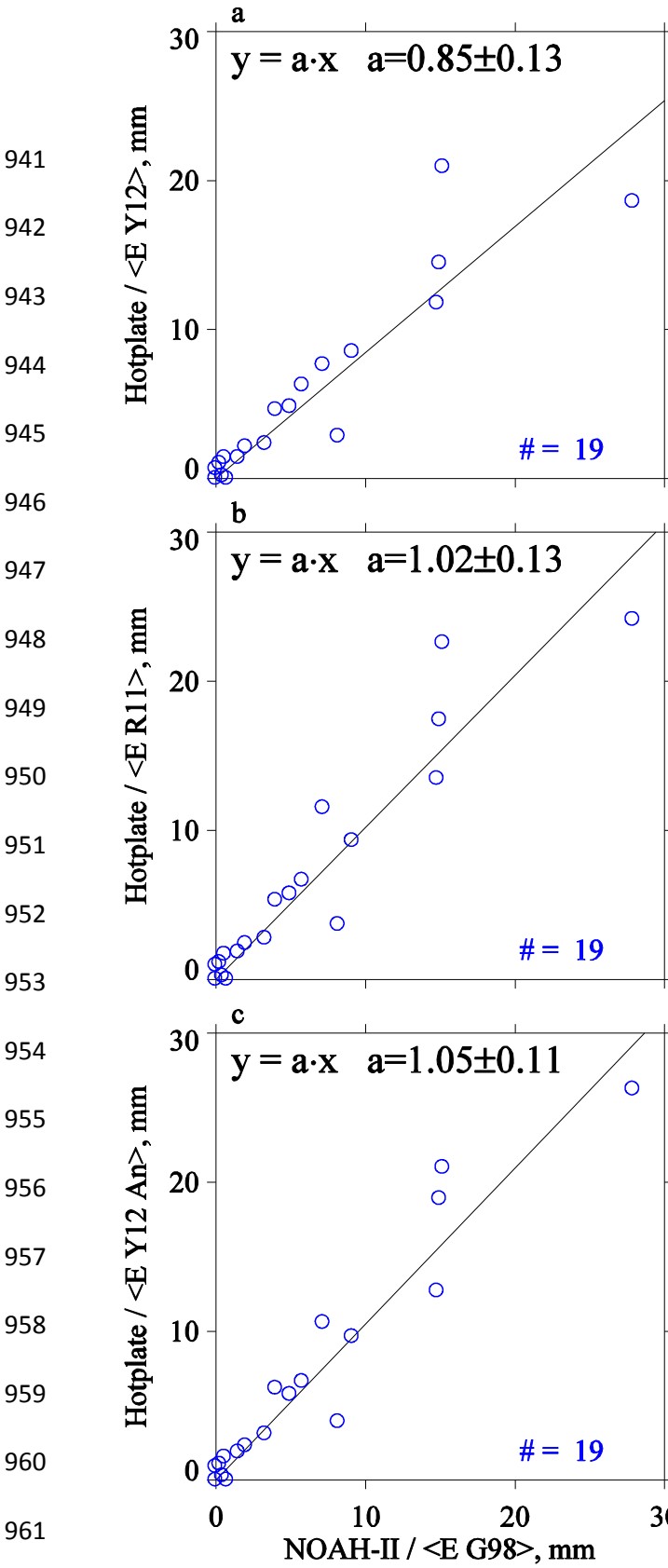

Figure 12 – UW hotplate and NOAH-II measurements of snow (liquid-equivalent

accumulations, not corrected for inefficient catch, divided by an event-averaged snow particle
catch efficiency) at OWL. An $f_2$ derived with the second of two methods discussed in section 3.8
was applied in the UW algorithm. Regression lines were forced through the origin and $y$
deviations (vertical departures of data from regression line) were used as the basis for the least
squares criterion of best fit (Young, 1962). The standard deviations on the fitted ratios
(confidence intervals) were derived using Student's t-distribution at the 95% level (Havilcek and
Crain, 1988).

**Table 1 – Hotplate Data Files**

| Recorded Variable [a], units | File UHP | File SHP | Symbol |
|---|:---:|:---:|:---:|
| Unix Time, s | ✓ | ✓ | |
| Liquid-equivalent Precipitation Rate, mm hr$^{-1}$ | ✓ | | $P_{YES}$ |
| Accumulated Liquid-equivalent Precipitation, mm | ✓ | | |
| Ambient Temperature, $^{o}$C | ✓ | ✓ | $T$ |
| Enclosure Temperature, $^{o}$C | ✓ | ✓ | |
| Wind Speed, m s$^{-1}$ | ✓ | | $U$ |
| Downwelling Shortwave Flux, W m$^{-2}$ | ✓ | ✓ | $SW$ |
| Longwave Radiation Measurement, W m$^{-2}$ | ✓ | ✓ | $M_{IR}$ |
| Barometric Pressure, hPa | ✓ | ✓ | $p$ [b] |
| Relative Humidity Sensor Temperature, $^{o}$C | ✓ | ✓ | |
| Relative Humidity, % | ✓ | ✓ | $RH$ |
| Top Plate Voltage, V | | ✓ | |
| Bottom Plate Voltage, V | | ✓ | |
| Top Plate Current, A | | ✓ | |
| Bottom Plate Current, A | | ✓ | |
| Top Plate Resistance, Ω | | ✓ | |
| Bottom Plate Resistance, Ω | | ✓ | |
| Top Plate Power, W | | ✓ | $Q_{top}$ |
| Bottom Plate Power, W | | ✓ | $Q_{bot}$ |
| Radiation Sensors' Temperature, $^{o}$C | | ✓ | |

[a] With the exception of Unix time, all recorded variables are 60-s running averages, sampled at 1 Hz
(YES, 2011)
[b] Although pressure is a recorded variable, the pressure used in the UW algorithm ($p_x$; section 3.7 and
Appendix) is the standard-atmosphere pressure at the altitude of the measurement

**Table 2 – Field Sites, Site Location, Vegetation at the Site, Gauge Location, Number of Events, and Event Type**

| Site Abbreviation Site Reference Hotplate | Site Location | Height of Vegetation, m AGL | Gauge Location at Site | Precipitation Events |
|---|---|---|---|---|
| GLE Wettlaufer (2013) UW | SE Wyoming 106.240 °W 41.3665 °N 3190 m MSL | 10 to 20 m | Hotplate: 27 m AGL on top deck of a meteorological tower [a] NOAH-II: 3 m AGL (clearing in conifer forest 80 m SE of tower) | 1 Snow |
| BTL Wettlaufer (2013) NCAR | SE Wyoming 106.975 °W 41.1558 °N 3010 m MSL | 10 to 20 m | Clearing in conifer forest Hotplate: 3 m AGL NOAH-II: 3 m AGL | 3 Snow |
| OWL Steenburgh et al. (2014) UW | NW New York 75.8771 °W 43.6245 °N 385 m MSL | 2 to 5 m | Clearing in deciduous brush and deciduous trees Hotplate: 1.7 m AGL [b] and 2.5 m AGL [c] NOAH-II: 2.5 m AGL | 4 Rain 19 Snow |

[a] This is the Brooklyn Lake, Wyoming AmeriFlux Tower; AmeriFlux is a network of sites that measure energy and trace-gas transfers.

[b] First 7 of 23 OWL precipitation events (Date < 20131217)

[c] Last 16 of 23 OWL precipitation events (Date > 20131217)

**Table 3 – Summary of Warm-cold Tests**

| Indoor Calibration (Warm-cold Tests) | | | |
|---|---|---|---|
| Year | Hotplate/Field Site | $T_h$, °C | $\gamma$ |
| 2012 | NCAR/BTL | 42.2 ± 7.4 [a] | 106. ± 15.1 [a] |
| 2012 - 2013 | UW/GLE | 52.2 ± 15.7 | 74.8 ± 18.1 |
| 2013 - 2015 | UW/OWL | 66.5 ± 7.8 | 57.8 ± 7.7 |

[a] Error limits derived by perturbing $Q_{top,w}$ (i.e., the value acquired in the warm test) by ± 0.5 W and repeating the analysis based on Eq. 7a - b.

**Table 4 – Summary of Drip Tests**

| Indoor Calibration (Drip Tests) | | | | |
|---|---|---|---|---|
| Year | Hotplate/Field Site | $P_{REF}$ vs $<P_{UW}>$ ratio [a] | $P_{REF}$ vs $<P_{YES}>$ ratio [a] | # [b] |
| 2012 | NCAR/BTL | $0.99 \pm 0.02$ | $0.81 \pm 0.03$ | 6 |
| 2012 - 2013 | UW/GLE | $1.00 \pm 0.06$ | $0.84 \pm 0.05$ | 6 |
| 2013 - 2015 | UW/OWL | $0.97 \pm 0.04$ | $0.79 \pm 0.03$ | 6 |

[a] Ratios were derived as the slope of a regression lines forced through the origin. The $x$ deviations (horizontal departures of data from regression line) were used as the basis for the least squares criterion of best fit (Young, 1962). Standard deviations on the fitted ratios (confidence intervals) were derived using Student's t-distribution at the 95% level (Havilcek and Crain, 1988).

[b] # = number of tests.

# Table 5 – Precipitation Events

| Precipitation Event | Start Date, YYYYMMDD UTC | Start Time, HH:MM UTC | End Time, HH:MM UTC | $\langle T \rangle$ [a], °C | $\langle T_{IB} \rangle$ [b], °C | $\langle U \rangle$ [c], m s$^{-1}$ | UW [d], mm | YES [d], mm | $\langle E\ Y12 \rangle$ [e] | $\langle E\ Y12\ An \rangle$ [f] | $\langle E\ R11 \rangle$ [g] | NOAH-II [d], mm | $\langle E\ G98 \rangle$ [h] |
|---|---|---|---|---|---|---|---|---|---|---|---|---|---|
| GLE-01 | 20120414 | 0:00 | 13:00 | -5.1 | -5.6 | 1.8 | 5.6 | 6.0 | 0.84 | NA [i] | NAP [j] | 7.3 | 0.89 |
| BTL-01 | 20120116 | 11:00 | 1:00 | -10.2 | -10.5 | 2.0 | 3.8 | 7.5 | 0.80 | NA | NAP | 8.0 | 0.87 |
| BTL-02 | 20120119 | 8:00 | 18:00 | -5.4 | -5.6 | 4.3 | 3.8 | 3.4 | 0.55 | NA | NAP | 1.0 | 0.60 |
| BTL-03 | 20120120 | 7:00 | 18:00 | -5.0 | -5.2 | 2.4 | 6.3 | 7.1 | 0.75 | NA | NAP | 8.2 | 0.83 |
| OWL-01 | 20131211 | 18:00 | 0:00 | -6.5 | -6.9 | 1.4 | 16.9 | 21.5 | 0.91 | 0.64 | 0.70 | 20.0 | 0.72 |
| OWL-02 | 20131212 | 0:00 | 6:00 | -10.9 | -10.7 | 0.4 | 1.0 | 3.5 | 0.99 | 0.94 | 0.90 | 0.3 | 0.96 |
| OWL-03 | 20131212 | 6:00 | 12:00 | -22.5 | -21.6 | 0.3 | 0.0 | 2.4 | 1.00 | 1.00 | 0.93 | 0.0 | 1.00 |
| OWL-04 | 20131212 | 18:00 | 0:00 | -9.8 | -10.5 | 1.3 | 0.2 | 1.2 | 0.90 | 0.53 | 0.67 | 0.3 | 0.57 |
| OWL-05 | 20131213 | 6:00 | 12:00 | -8.4 | -9.0 | 0.9 | 13.9 | 18.5 | 0.96 | 0.73 | 0.80 | 12.1 | 0.81 |
| OWL-06 | 20131215 | 19:45 | 0:00 | -6.3 | -6.9 | 0.9 | 1.3 | 2.9 | 0.94 | 0.85 | 0.78 | 0.5 | 0.88 |
| OWL-07 | 20131216 | 0:00 | 6:00 | -6.4 | -7.3 | 0.3 | 20.9 | 27.8 | 1.00 | 1.00 | 0.93 | 15.2 | 1.00 |
| OWL-08 | 20131218 | 18:00 | 0:00 | -3.3 | -3.7 | 1.5 | 10.4 | 13.2 | 0.88 | 0.82 | 0.77 | 12.9 | 0.87 |
| OWL-09 | 20140106 | 18:00 | 0:00 | -6.2 | -6.5 | 3.9 | 4.5 | 4.6 | 0.58 | 0.42 | 0.39 | 2.5 | 0.35 |
| OWL-10 | 20140107 | 0:00 | 6:00 | -14.4 | -14.7 | 3.8 | 0.4 | 0.3 | 0.60 | 0.44 | 0.42 | 0.0 | 0.38 |
| OWL-11 | 20140107 | 6:00 | 12:00 | -17.3 | -17.4 | 3.0 | 1.9 | 3.3 | 0.67 | 0.49 | 0.52 | 4.0 | 0.49 |
| OWL-12 | 20140107 | 18:00 | 0:00 | -17.5 | -17.9 | 3.9 | 0.0 | 0.0 | 0.58 | 0.42 | 0.39 | 0.3 | 0.35 |
| OWL-13 | 20140111 | 15:45 | 21:00 | 7.2 | 6.6 | 1.1 | 11.9 | 14.1 | 1.00 | 1.00 | 1.00 | 13.2 | 1.00 |
| OWL-14 | 20140111 | 23:00 | 2:00 | 6.4 | 5.8 | 2.4 | 3.9 | 3.9 | 1.00 | 1.00 | 1.00 | 3.5 | 1.00 |
| OWL-15 | 20140114 | 0:00 | 12:00 | 3.7 | 2.9 | 0.9 | 16.1 | 19.9 | 1.00 | 1.00 | 1.00 | 15.3 | 1.00 |
| OWL-16 | 20140114 | 12:45 | 15:45 | 4.1 | 3.6 | 1.2 | 2.6 | 3.1 | 1.00 | 1.00 | 1.00 | 1.8 | 1.00 |
| OWL-17 | 20140119 | 0:00 | 12:00 | -7.5 | -7.8 | 1.3 | 1.9 | 3.3 | 0.91 | 0.84 | 0.81 | 1.8 | 0.88 |
| OWL-18 | 20140119 | 18:00 | 0:00 | -5.7 | -6.5 | 1.9 | 1.9 | 2.6 | 0.81 | 0.61 | 0.69 | 2.3 | 0.69 |
| OWL-19 | 20140120 | 0:00 | 2:00 | -3.7 | -4.1 | 3.1 | 0.9 | 1.0 | 0.67 | 0.50 | 0.52 | 0.8 | 0.51 |
| OWL-20 | 20140120 | 2:00 | 12:00 | -3.5 | -3.9 | 2.1 | 3.8 | 4.8 | 0.78 | 0.66 | 0.66 | 3.5 | 0.72 |
| OWL-21 | 20140127 | 18:00 | 0:00 | -11.5 | -12.3 | 1.7 | 4.0 | 6.6 | 0.86 | 0.65 | 0.75 | 2.8 | 0.69 |
| OWL-22 | 20140128 | 0:00 | 6:00 | -15.0 | -15.6 | 0.4 | 6.3 | 10.3 | 1.00 | 0.95 | 0.94 | 5.6 | 0.96 |
| OWL-23 | 20140128 | 6:00 | 12:00 | -16.5 | -17.3 | 0.9 | 8.0 | 12.1 | 0.94 | 0.83 | 0.86 | 8.1 | 0.88 |

[a] Event-averaged ambient temperature

[b] Event-averaged ice bulb temperature

[c] Event-averaged hotplate $U$

[d] Liquid-equivalent precipitation amount not corrected for inefficient catch (UW values are computed with an $f_2$ derived with the second of two methods discussed in section 3.8)

[e] Event-averaged snow particle catch efficiency derived using Y12 and the hotplate $U$

[f] Event-averaged snow particle catch efficiency derived using Y12 and the anemometer $U$

[g] Event-averaged snow particle catch efficiency derived using R11 and hotplate $U$ adjusted to 10 m AGL

[h] Event-averaged snow particle catch efficiency derived using G98 and hotplate $U$ (GLE and BTL) or anemometer $U$ (GLE)

[i] NA = not available

[j] NAP = not applicable

1    **Table 6 – Summary of Fitted *Nu-Re* Coefficients**

| Field Calibration (*Nu* - *Re* Coefficients) | | | |
|---|---|---|---|
| Hotplate/Field Site | $\gamma$ | $\alpha$ | $\beta$ |
| NCAR/BTL [a] | 86.2 | 0.126 | 0.781 |
| UW/GLE [b] | 49.1 | 0.130 | 0.771 |
| UW/OWL [c] | 45.6 | 0.172 | 0.713 |

9    [a] NCAR hotplate; measurement interval 20120118 23:00 UTC  to 20120119 5:00 UTC

10   [b] UW hotplate; measurement interval 20120402 04:00 UTC to 20120402 09:00 UTC

11   [c] UW hotplate; measurement interval 20140107 18:00 UTC to 20140108 08:00 UTC

