# Peer review of "Hotplate Precipitation Gauge Calibrations and Field Measurements"

_Atmospheric Measurement Techniques, 2017_

## Short Comment (SC1) · 7 Sep 2017

R. Rasmussen

rasmus@ucar.edu

1. Area incorrect: I agree that the area in my original paper is incorrect. We had a number of prior designs of the hotplate and this is probably left over from one of those sensor heads. Boudala et al. (2014) also found this. Not sure how I missed it in proof reading. However, I never expected the factor to be perfect, so I fully expected it to need calibration, so at the time I was mainly worried about the hotplate working and not the exact value of the theoretical factor. I think the algorithm worked well for the data we collected, I never thought that the Yankee algorithm would have the same factor (in fact, I don't think it does). The reasoning used on page 12 is correct, however, I expected the calibration factor not to be exactly equal to the terms which is why I called it a "calibration" factor and not a constant. 2. The snow particle collection efficiency is

the most uncertain part of the original hotplate algorithm. It worked well for the data I had a Marshall, but I was not sure it would work as well elsewhere. The recent WMO solid precipitation experiment evaluated three hotplates, and they performed well, in fact, they had the lowest RMSE of all the snow gauges tested. I attribute this to the aerodynamic profile, making the dependence of snow particle type less than weighing gauges. This report will be coming out before January 1, 2018. We did spend a lot of time worrying about the level that the wind speed was taken from, so your discussion on this is useful. However, the Yankee algorithm uses the wind from the hotplate itself, so the raw data is from 2 meters. 3. Do you have photos of the sites? Are they flat? The hotplate will be biased if the wind is not horizontal or there is upstream blocking. What I would also like to see is an uncertainty analysis of the truth gauge (ETI) as compared to the hotplate data. 4. The paper is quite detailed. If possible, I would try to focus on less of the details of the testing and more on the results. How well is the hotplate performing given the current algorithm? How much accuracy is gained by including radiation (%)? 5. I also tried to calculate the Nusselt number but never got a satisfactory comparison to a flat plat. I assumed this was due to the addition of the ridges. As a result I left this out of the original paper. 6. One of the major findings in running a hotplate that I found is that its performance depends on whether it is outdoors or indoors. Thus, the outdoor turbulence, in my experience, makes the hotplate perform a lot different than in a wind tunnel. In the wind tunnel, the data are very clean and everything works as you would expect. Outdoors, the turbulence on the bottom plate and upper plates impact the cooling differently, causing me to take a 5 minute average before initiating accumulation. 7. The final comparison to outdoor data (pages21-23) is confusing to me (and I expect an independent reader). The SPICE evaluation of the YES hotplate suggests that it is 10% high, yet you find it 10% or more low. I am not sure if this is due to using the wrong level for the wind or a different wind correction algorithm. If YES has a different catch efficiency algorithm than I do I don't know where they got it from as they did not do any outdoor testing in comparison to a truth gauge. I think it might be useful to have a discussion on these data next time

you are in Boulder or I am in Laramie. We could also do a conference call, but it might take some time to figure this out. 8. Conclusions: I think you can state that the area is incorrect, but, again, the factor was not stated as a constant but a calibration factor. I think this point is over-stated and was already made by Boudala et al. (2014). 9. I would state the main conclusions in the final section. What I want to hear about is: 1. Is a radiation correction important to the hotplate?, 2. How well does the Yankee hotplate do compared to field observations? The wind speed in the Yankee hotplate is from the hotplate itself, so it self consistent. The fact that the unit has performed well for me and SPICE over the past 5 years suggests that the algorithm if fundamentally sound. 3. Any suggestions on how to improve the catch efficiency? How variable is it from storm to storm? I think a histogram plot of the delta precipitation as compared to a truth gauge as I did in my 2011 paper would be very useful. Are the differences biased one direction or another? Do we need to take into account snow type, for instance. Are there are observations of snow type? The largest discrepancy I saw was for graupel. Was there graupel particles observed during OWLS?

---

## Author Comment (AC1) · 19 Sep 2017

We thank Professor Roy Rasmussen for commenting on the manuscript.

1) Area incorrect: I agree that the area in my original paper is incorrect. We had a number of prior designs of the hotplate and this is probably left over from one of those sensor heads. Boudala et al. (2014) also found this.

We referenced Boudala et al. (2014). In our opinion, explaining how the conversion factor is handled in Rasmussen et al. (2011) (R11) requires consideration of the area effect, and the 20% lowering described here in R11:

"In practice, this conversion factor was 20% lower because of the imperfect heat transfer from the precipitation to the hot plate (losses to the air, e.g.). The actual conversion rates were determined by comparing the predicted precipitation rate from the hotplate to the measured rates from a GEONOR precipitation gauge in the Double-Fence Intercomparison Reference (DFIR) shield (see discussion below)."

Because the area effect and the 20% lowering are linked, we discuss them together, and thus we repeat the finding of Boudala et al. (2014).

Not sure how I missed it in proof reading. However, I never expected the factor to be perfect, so I fully expected it to need calibration, so at the time I was mainly worried about the hotplate working and not the exact value of the theoretical factor. I think the algorithm worked well for the data we collected, I never thought that the Yankee algorithm would have the same factor (in fact, I don't think it does).

As we stated L268-L274, we **assumed** that the YES algorithm has incorporated R11's surface area and R11's distinction between the theoretical and actual conversion factors.

The reasoning used on page 12 is correct, however, I expected the calibration factor not to be exactly equal to the terms which is why I called it a "calibration" factor and not a constant.

Agreed, it's not a constant. To be consistent with R11, we used "energy conversion factor."

2) The snow particle collection efficiency is the most uncertain part of the original hotplate algorithm. It worked well for the data I had a Marshall, but I was not sure it would work as well elsewhere. The recent WMO solid precipitation experiment evaluated three hotplates, and they performed well, in fact, they had the lowest RMSE of all the snow gauges tested. I attribute this to the aerodynamic profile, making the dependence of snow particle type less than weighing gauges. This report will be coming out before January 1, 2018. We did spend a lot of time worrying about the level that the wind speed was taken from, so your discussion on this is useful. However, the Yankee algorithm uses the wind from the hotplate itself, so the raw data is from 2 meters.

As we explain on L295-L298, we used information conveyed in a personal communication from YES to formulate the snow particle catch efficiency function. In the manuscript, this is indicated "Y12." From that personal communication, the Y12 efficiency function is based on the hotplate-derived wind speed. It is our understanding that there is **no** adjustment of the hotplate-derived wind speed that is input to the Y12 snow particle catch efficiency function.

3) Do you have photos of the sites?

Yes, photographs are available. These are in the references provided in the first column of Table 2.

Are they flat?

No, however, our calculation of the sensible energy output term (the dominant term in the power budget) is based on site-specific calibrations. This is discussed L418-L438.

The hotplate will be biased if the wind is not horizontal or there is upstream blocking. What I would also like to see is an uncertainty analysis of the truth gauge (ETI) as compared to the hotplate data.

Figure 12b shows that when we apply the catch efficiency function recommended by R11 (E R11) our result compares well with weighing gauge measurements that were also corrected for inefficient catch. The Figure 12b is a scatterplot of the 19 analyzed OWL snow events in our data set. Development of an uncertainty analysis for the NOAH-II gauge (aka, ETI gauge) would require significantly more information. For example, as shown in Kochendorfer et al. (2017), this can involve hundreds of concurrent measurements from a DFIR and the gauge under test.

Kochendorfer, J., Rasmussen, R., Wolff, M., Baker, B., Hall, M. E., Meyers, T., Landolt, S., Jachcik, A., Isaksen, K., Brækkan, R., and Leeper, R.: The quantification and correction of wind-induced precipitation measurement errors, Hydrol. Earth Syst. Sci., 21, 1973-1989, https://doi.org/10.5194/hess-21-1973-2017, 2017

4) The paper is quite detailed.  If possible, I would try to focus on less of the details of the testing and more on the results. How well is the hotplate performing given the current algorithm?

As stated in the abstract, and in the conclusion, the manuscript has two objectives: 1) incorporation of the radiation measurements into the power budget, and thus into what we refer to as the Wyoming Algorithm, and 2) reconciling R11, and possibly what YES is doing in their real-time processor, with the Wyoming Algorithm.  In our opinion, the detail provided is necessary for meeting these objectives.

Results presented in Figure 11 (rain) and in Figure 12b (snow) address your second point.

How much accuracy is gained by including radiation (%)?

> The effect of radiation is addressed L476 to L484.

5) I also tried to calculate the Nusselt number but never got a satisfactory comparison to a flat plat. I assumed this was due to the addition of the ridges. As a result I left this out of the original paper.

> This is discussed in response to your point #3. We feel that the approach we have developed is useful because it allows incorporation of the sensible energy term into power budget (Equation 3, manuscript).

6) One of the major findings in running a hotplate that I found is that its performance depends on whether it is outdoors or indoors. Thus, the outdoor turbulence, in my experience, makes the hotplate perform a lot different than in a wind tunnel. In the wind tunnel, the data are very clean and everything works as you would expect. Outdoors, the turbulence on the bottom plate and upper plates impact the cooling differently, causing me to take a 5 minute average before initiating accumulation.

> We have not used a wind tunnel. Just to be clear, our indoor testing was in the hangar or in the laboratory. There was ventilation, but the observed wind speeds were below the detection threshold of the hotplate (0.1 m/s).

7) The final comparison to outdoor data (pages21-23) is confusing to me (and I expect an independent reader). The SPICE evaluation of the YES hotplate suggests that it is 10% high, yet you find it 10% or more low. I am not sure if this is due to using the wrong level for the wind or a different wind correction algorithm. If YES has a different catch efficiency algorithm than I do I don't know where they got it from as they did not do any outdoor testing in comparison to a truth gauge. I think it might be useful to have a discussion on these data next time you are in Boulder or I am in Laramie. We could also do a conference call, but it might take some time to figure this out.

I presume, what you are comparing, in SPICE, are the values produced by the internal YES algorithm. In the manuscript, these are referred to as hotplate-derived accumulations. The figure here shows the 19 analyzed OWL snow events with hotplate-derived accumulations (ordinate) vs NOAH-II values (abscissa). The collection efficiencies applied here are event averages and these were divided into accumulations that were not corrected for inefficient catch. Data presented in the figure are available in our Table 5. The slope of the fit line is 1.1, consistent with the "10% high" you mention, but there is large statistical uncertainty due to the relatively small number of points and scatter in the measurements.

[Figure]

8) Conclusions: I think you can state that the area is incorrect, but, again, the factor was not stated as a constant but a calibration factor. I think this point is over-stated and was already made by Boudala et al. (2014).

Please see our related comments above. We feel our repeating of R11's distinction between a theoretical and an actual energy conversion factor is appropriate.

9) I would state the main conclusions in the final section.  What I want to hear about is:

a) Is a radiation correction important to the hotplate?

This is addressed in the body of the manuscript.  We agree that the information should be restated in the conclusions.

b) How well does the Yankee hotplate do compared to field observations? The wind speed in the Yankee hotplate is from the hotplate itself, so it self consistent. The fact that the unit has performed well for me and SPICE over the past 5 years suggests that the algorithm if fundamentally sound.

At OWL, the YES-derived wind speeds are biased low relative to the anemometer-derived wind speeds. This is discussed L491 to L516. We speculate that the hotplate-derived accumulations in Figure 12a are statistically smaller than NOAH-II values because 1) the Y12 catch efficiency function does not apply a height-adjusted U, or 2) because of the hotplate underestimate of U, compared to the anemometer. We conclude that untangling these two possibilities is beyond the scope of the study.

c) Any suggestions on how to improve the catch efficiency? How variable is it from storm to storm?

Event-averaged catch efficiency values are in Table 5.

I think a histogram plot of the delta precipitation as compared to a truth gauge as I did in my 2011 paper would be very useful. Are the differences biased one direction or another? Do we need to take into account snow type, for instance.

There are 19 analyzed OWL snow events in our data set.  In our view, the best way to present these is in a scatterplot. This is done in Figure 12a – c. Meaningful consideration of the additional effects (e.g., wind direction) will require more data.

Are there are observations of snow type? The largest discrepancy I saw was for graupel.  Was there graupel particles observed during OWLS?

In our data set there are three OWL snow events (out of 19) with either riming or graupel mentioned in the notes taken by Leah Campbell and Jim Steenburgh. In two of these (OWL-05 and OWL-20; Table 5) the UW hotplate accumulation (not corrected for inefficient catch) divided by $<E R11>$, is larger that the NOAH-II accumulation (not corrected for inefficient catch) divided by $<E G98>$. This can be verified using values in our Table 5.

---

## Referee Comment (RC1) · C. Genthon (Referee) · 25 Sep 2017

This paper demonstrates that an original algorithm used in the Total Precipitation sensor TPS-3100 also called ÂńÂăhotplateÂăÂż precipitometer to derive quantities of precipitation has some errors that can be corrected, and has shortcomings that can be reduced. This is based on both theoretical and experimental approaches. The Hotplate concept uses an original method to quantify precipitation: the differential energy needed to keep 2 overlaid hot plates at constant temperature, one facing up and the other facing down, includes the contribution (latent heat) needed to evaporate any condensed water falling on the upper plate (but obviously not affecting the lower plate facing down). There are many other terms to account for though. Both plates loose sensible energy and the loss rate is related to wind speed and ambient temperature.

[Figure]

However the 2 plates are equally affected so the differential is 0. On the other hand, the radiation balance of the 2 plates is not necessary the same and the differential must be accounted for. The issue of the catch efficiency of the instrument is also raised, particularly for solid particles.

The hotplate concept is very attractive for a number of reasons listed in the paper, including the fact that it is immune to frost deposition and snow/frost clogging, thus particularly appropriate in cold regions. Yet, although the instrument is not new (marketed by YES since the early 2000's), it does not seem to be so widely used (the paper mentions that "70 Yankee Environmental Systems (YES) hotplate precipitation gauges have been purchased by researchers and operational meteorologists" which is not a lot). This may (or may not, price is also an issue of course) sign some dissatisfaction with the results obtained, and an improvement of the algorithm may contribute popularize the instrument.

If the approach and methods appear sound, there is one potential major shortfall with the paper: it is not obvious if and how the revised algorithm can be implemented in an existing units. This is admittedly a technical issue but this is "atmospheric measurement techniques" so one expect technical issues to be addressed.

The paper mentions that the hotplate outputs data in 2 files, one (UHP) which "is provided to all YES customers" and the other (SHP) with no indication how it can be accessed. Of course the later has the data needed to design and use a new algorithm but there is no reference to the second file in the YES Hotplate documentation as of 2011. YES no longer markets the instrument and no longer provides any information on its web site. In such a paper, one should not have to bet on obtaining "private communication" from a manufacturer which has terminated production, to learn how to access the data needed to implement the improved algorithm. I believe that the paper has very limited significance and is not acceptable for publication in its current form unless this information,and all information necessary to implement the new algorithm, is clearly provided. On the other hand, it definitively ranks publication status if the information is given so that the reader can implemented the new algorithm, and provided the various issues bellow are addressed, some of which are fairly serious though.

The introduction (lines 33 – 37) states that 2 types of instrumentations to measure precipitation have been developed: the capture and optical gauges. This ignores radars which are powerful tools to measure and even profile precipitation. Because of ground clutter and vertical resolution, radars do admittedly not measure precipitation right at the very surface but because they can profile vertically it may be checked whether precipitation rates vary or not as it reaches closer to the surface. Radars do not "obstruct the wind and deflects falling particles in the measurement zone" (line 38). There is no "clogging with snow" with radars (line 47).

Equation (3) (lines 113 – 120) describes the improvement of the algorithm by the authors by taking into account the solar and thermal radiation contributions the the heat balance of the upper plate. Except for the latent heat term, one would expect a similar expression for the lower plate but this is not explicit. Because the hotplate is immune from snow and frost related problems that affect other sensors, it is expected particularly useful in cold snowy regions but then comes in the short wave power input reflected by the surface. If the sensor only measures the downwelling solar radiation, how is the reflected part factored in the lower plate energy balance?

Line 103: the sensible term is a function of U and T. Figure 1 shows that the temperature sensor is very poorly shaded from solar radiation reflected by the surface. Over a snow covered surface this is a likely major problem. The authors should evaluate the impact on precipitation estimation over snow, possibly bring in an empirical correction?

Then, according to line 202, the upwelling IR is also estimated from this temperature supposed to be the ambient air temperature, but probably largely overestimated over snow.

Line 3176: Define AGL (presumably Above Ground Level)

Line 376: How would a "violation of the steady state assumption" could explain the delay? Would this have to do with thermal inertia?

Lines 422 – 423: This is not clear: the authors use data derived for a flow perpendicular to the plates to determine parameters for a flow parallel to the plates? Does this make sense? Can you clarify?

Table 1: "Hotplate data files". This table is misleading has long as access to SHP data is not explicit.

Lines 527-530: A synthesis table describing the various algorithms referenced in the paper could be useful. Only here does one clearly realize that none of the above algo discussion applies to the commercial (YES) one. The average hotplate user, presumably a target reader, probably got his/her instrument from YES and expects he/she will be able to improve his/her instrument with the new algo. It is not necessarily obvious from the beginning that R11 is not YES algo and that the authors describe an algo which objectively improve over R11 but not necessarily over the commercial units. Improvement over the commercial algo is verified only in the end. This should be clearly stated from the very beginning.

---

## Author Comment (AC2) · 6 Oct 2017

We thank Professor Genthon for reviewing the manuscript.

This paper demonstrates that an original algorithm used in the Total Precipitation sensor TPS-3100 also called "hotplate" to derive quantities of precipitation has some errors that can be corrected, and has shortcomings that can be reduced. This is based on both theoretical and experimental approaches. The Hotplate concept uses an original method to quantify precipitation: the differential energy needed to keep 2 overlaid hotplates at constant temperature, one facing up and the other facing down, includes the contribution (latent heat) needed to evaporate any condensed water falling on the upper plate (but obviously not affecting the lower plate facing down). There are many other terms to account for though.Both plates loose sensible energy and the loss rate is related to wind speed and ambient temperature.

However the 2 plates are equally affected so the differential is 0. On the other hand, the radiation balance of the 2 plates is not necessary the same and the differential must be accounted for. The issue of the catch efficiency of the instrument is also raised, particularly for solid particles.

The hotplate concept is very attractive for a number of reasons listed in the paper, including the fact that it is immune to frost deposition and snow/frost clogging, thus particularly appropriate in cold regions. Yet, although the instrument is not new (marketed by YES since the early 2000's), it does not seem to be so widely used (the paper mentions that "70 Yankee Environmental Systems (YES) hotplate precipitation gauges have been purchased by researchers and operational meteorologists" which is not a lot). This may (or may not, price is also an issue of course) sign some dissatisfaction with the results obtained, and an improvement of the algorithm may contribute popularize the instrument.

If the approach and methods appear sound, there is one potential major shortfall with the paper: it is not obvious if and how the revised algorithm can be implemented in an existing units. This is admittedly a technical issue but this is "atmospheric measurement techniques" so one expect technical issues to be addressed.

The paper mentions that the hotplate outputs data in 2 files, one (UHP) which "is provided to all YES customers" and the other (SHP) with no indication how it can be accessed. Of course the later has the data needed to design and use a new algorithm but there is no reference to the second file in the YES Hotplate documentation as of 2011.

We asked Roy Rasmussen at the National Center for Atmospheric Research (NCAR) for access to the SHP data and were granted access after signing a legal agreement that protects the real time processor from piracy. We have presented analyses of SHP data in two theses, and now, in this manuscript in AMTD.

Here is how we will revise section 2.1:

2.1 - Hotplate Data Files

The hotplate outputs data to two files. The previously discussed $Q_{top}$ and $Q_{bot}$ are two of several recorded variables and both of these are essential for the analysis described here. One of the files is known as the UHP or "user" hotplate file. The UHP file is provided to all YES customers. The second file is the SHP or "sensor" file. The SHP file is proprietary but we were granted access to it by the National Center for Atmospheric Research (NCAR; Boulder, CO). Table 1 has the list of all recorded variables and how some of these are symbolized. A complete list of variables (measured and computed), and constants, is provided in the Appendix. With the exception of Unix time, all variables in Table 1 are provided as 60-s running averages, sampled at 1 Hz (YES, 2011).

YES no longer markets the instrument and no longer provides any information on its web site. In such a paper, one should not have to bet on obtaining "private communication" from a manufacturer which has terminated production, to learn how to access the data needed to implement the improved algorithm.

As we understand it, termination of production/marketing is temporary and occurred because of a legal argument between UCAR and YES. Currently, YES is servicing our hotplate.

I believe that the paper has very limited significance and is not acceptable
for publication in its current form unless this information, and all information
necessary to implement the new algorithm, is clearly provided.

We want to probe your comment in the last sentence. Are you saying that more
information about the UW Algorithm is needed? It is our opinion that we provided
enough for a user to implement the UW algorithm, provided they have access to the
SHP data. Also, please see our previous statement that we will revise section 2.1 to
include discussion of how the SHP data was accessed.

On the other hand, it definitively ranks publication status if the information is given so that the reader can implemented the new algorithm, and provided the various issues bellow are addressed, some of which are fairly serious though.

1) The introduction (lines 33 – 37) states that 2 types of instrumentations to measure precipitation have been developed: the capture and optical gauges.  This ignores radars  which are powerful tools to measure and even profile precipitation.

Because of ground clutter and vertical resolution, radars do admittedly not measure precipitation right at the very surface but because they can profile vertically it may be checked whether precipitation rates vary or not as it reaches closer to the surface. Radars do not "obstruct the wind and deflects falling particles in the measurement zone" (line 38). There is no "clogging with snow" with radars (line 47).

In the US National Weather Service (NWS) network, a radar-derived snow amount is dependent on gauge measurements made simultaneous with the radar measurements. A published example of this is the NWS gauge-radar network data analyzed by Martinaitis et al. (2015). In that network, and presumably others, the gauge-calibrated radar-derived estimates of snow can be significantly biased.

Factors contributing to this are gauge clogging by snowfall, postevent thaw of snow, and underestimation of snow because wind speed is either not used to correct for gauge undercatch or it is unavailable. Martinaitis et al. (2015) state that "..The accuracy of hourly radar-derived QPE values of winter precipitation is unknown..".

Given this caveat, the dependence of radar precipitation estimates on gauge measurements, and the factors you mention (vertical structure, Earth curvature, ray ducting, ground clutter, and etc.), we have opted to not mention that radars can be used to derive snowfall amounts.

Martinaitis, S.M., S.B.Cocks, Y.Qi, B.T.Kaney, J.Zhang, and K.Howard, Understanding
winter precipitation impacts on automated gauge observations within a real-time
system, J. Hydrometeor., 16, 2345-2363, https://doi.org/10.1175/JHM-D-15-
0020.1, 2015

Equation (3) (lines 113 – 120) describes the improvement of the algorithm by the authors by taking into account the solar and thermal radiation contributions the heat  balance of the upper plate. Except for the latent heat term, one would expect a similar expression for the lower plate but this is not explicit. Because the hotplate is immune from snow and frost related problems that affect other sensors, it is expected particularly useful in cold snowy regions but then comes in the short wave power input reflected by the surface. If the sensor only measures the downwelling solar radiation,  how is the reflected part factored in the lower plate energy balance?

Rasmussen et al. (2011) factored bottom plate power and ambient temperature, both measured at 2 m AGL, into a function that was regressed against wind speeds measured at 10 m AGL. This is evident in section 3a of their paper. Because the bottom plate power is affected by wind, snow-reflected solar, and longwave exchange, they have accounted for (implicitly) the effect you are referring to.

We hope that Professor Rasmussen and Mark Beaubien at YES can comment more on this.

Line 103: the sensible term is a function of U and T. Figure 1 shows that the temperature sensor is very poorly shaded from solar radiation reflected by the surface. Over a  snow covered surface this is a likely major problem.  The authors should evaluate the  impact on precipitation estimation over snow, possibly bring in an empirical correction?

In Table R-1 we estimate this error for nonprecipitating conditions. Here the type–1 accumulations are evaluated using the UW algorithm and the type-2 accumulations are evaluated by substituting into the UW algorithm coincident temperature measurements made within a shielded temperature sensor (Vaisala WXT520). Results are presented for four days. The reported liquid equivalent accumulations are derived by integrating liquid equivalent rates from 16 to 20 UTC (10 to 14 LT). The table shows that positive bias in type-1, relative to type-2, is no larger than 1.4 liquid equivalent mm in four hours. Further, it needs to be acknowledged that solar is attenuated when precipitation is occurring, and thus the effect of sensor heating on precipitation measurements is smaller than demonstrated in Table R-1. In our opinion, more analysis is needed to explain and quantify this effect for both precipitating and nonprecipitating periods.  Thus, we feel that the reviewer's recommendation is beyond the scope of the manuscript.

Table R-1 - Accumulations derived for nonprecipitating four-hour daytime cases; 16 to 20 Universal Coordinated Time (10 to 14 LT). Measurements are from the UW hotplate operated at the Medicine Bow Mountain Site described in Zelasko (2017).

| UTC Year Month Day | 20170128 | 20170129 | 20170130 | 20170131 |
|---|---|---|---|---|
| Type-1 [a] Accumulation, mm [c] | 1.7 | 1.7 | 1.8 | 1.4 |
| Type-2 [b] Accumulation, mm [c] | 0.5 | 0.7 | 0.4 | 0.1 |

[a] Type-1 are liquid-equivalent four-hour accumulations derived from UW hotplate measurements using the UW processing algorithm

[b] Type-2 are liquid-equivalent four-hour accumulations derived from UW hotplate measurements using the UW processing algorithm with Vaisala WXT520 temperatures substituted for YES temperatures

[c] The accumulation unit is liquid-equivalent depth over four hours (16 to 20 UTC)

Zelasko, N., Orographic Precipitation in Southeastern Wyoming, MS Thesis, Department of Atmospheric Science, University of Wyoming, 2017

Then, according to line 202, the upwelling IR is also estimated from this temperature supposed to be the ambient air temperature, but probably largely overestimated over snow.

The term is overestimated by sensor heating (Wolfe and Snider, 2012, their section

2c), however, the term contributes is less than 3 % to the power budget. This is demonstrated below (Figure R-1) for the same nonprecipitating daytime cases as in

Table R-1. Furthermore, during precipitation the solar is attenuated, and thus the magnitude of the error is less than that for these nonprecipitating cases.

Wolfe, J.P., and J.R.Snider, A Relationship between Reflectivity and Snow Rate for a High-Altitude S-Band Radar, J. Appl. Meteor. Climatol., 51, 1111–1128,

2012

[Figure]

Figure R-1 – Left) Time sequence of $A_h \cdot \varepsilon_h \cdot \sigma \cdot T^4$. Also plotted is the sum of all input terms in the power budget. Right) Ratio $A_h \cdot \varepsilon_h \cdot \sigma \cdot T^4$ divided by all inputs. Results are for the nonprecipitating daytime cases. The field site is described in (Zelasko, 2017).

Line 3176: Define AGL (presumably Above Ground Level)

In the revision, we will define AGL.

Line 376: How would a "violation of the steady state assumption" could explain the delay? Would this have to do with thermal inertia?

We do not see a "delay", rather a "slowed response". Two things are contributing to this.

1) Each 1-s sample is a 60 s running average (see L131 and L977 in the manuscript), and

2) thermal inertia. Better understanding of the thermal inertia will require power measurements that are not 60-s averaged. We have asked YES for this.

Lines 422 – 423: This is not clear: the authors use data derived for a flow perpendicular to the plates to determine parameters for a flow parallel to the plates? Does this make sense? Can you clarify?

Hansen and Webb (1992) is the only publication, we are aware of, that reports experimentally-derived values of $\alpha$ and $\beta$ for a circular plate with concentric rings.

And yes, the airflow in Hansen and Webb (1992) was perpendicular (normal to) the plate surface. We did *not* mean to imply that the values of $\alpha$ and $\beta$ determined by

Hansen and Webb (1992) were used in our analysis.

Here are revisions that should clarify the sentence on L420:

This result is based on UW hotplate measurements, acquired at the GLE site, and formulas developed in section 3.6.

Revision for L227 to L229.

In this section, we develop a relationship between *Nu* and *Re* based on measurements recorded in the field when precipitation was not occurring; in a later section we show how that relationship is applied in the new algorithm.

Table 1: "Hotplate data files". This table is misleading has long as access to
SHP data is not explicit.

We apologize for not stating how we acquired the SHP data. As discussed above,
we plan to revise section 2.1 so this is clear.

Lines 527-530: A synthesis table describing the various algorithms
referenced in the paper could be useful. Only here does one clearly realize that
none of the above algo discussion applies to the commercial (YES) one. The
average hotplate user, presumably a target reader, probably got his/her
instrument from YES and expects he/she will be able to improve his/her instrument
with the new algo. It is not necessarily obvious from the beginning that R11 is not
YES algo and that the authors describe an algo which objectively improve over R11
but not necessarily over the commercial units. Improvement over the commercial
algo is verified only in the end. This should be clearly stated from the very
beginning.

On L268-L274, near the beginning of the manuscript, we stated that we **assumed**
that the YES algorithm has incorporated R11's surface area and R11's distinction
between the theoretical and actual conversion factors. At present, we do not know if
these effects are actually incorporated into the YES algorithm.

We do not think that a synthesis table is necessary. If the reviewer is insistent, we
think it should go toward the end of the manuscript.

---

## Referee Comment (RC2) · J. Kochendorfer (Referee) · 9 Oct 2017

**Overview and Summary**

"Hotplate precipitation gauge calibrations and field measurements" describes theoretical and observational work focused on a novel precipitation gauge. The hotplate precipitation gauge measures precipitation by recording the amount of energy that is consumed by heating, melting, and evaporating precipitation captured on an upward-facing heated plate. The manuscript includes improvements to algorithms used to estimate the wind-induced undercatch of the hotplate, the conversion of energy to evaporation, and the effects of radiation, temperature, and wind on the energy balance and precipitation rate of the sensor. These algorithms are developed and tested using field and laboratory measurements.

Although the hotplate is not widely used, it is a unique, low-maintenance sensor capable of measuring all forms of precipitation in areas where power is available. Based on the hotplate's good performance in SPICE, there may be renewed interest in this technology when the SPICE results become widely available. The refinements and testing of the hotplate described in the present manuscript are therefore both valuable and timely.

Some general comments below indicate areas where there is room for improvement. The manuscript is generally well written, but typos and other more specific suggestions are documented in the specific comments.

**General Comments**

The description of the hotplates and algorithms discussed in the Introduction and Methods sections should be augmented to clearly state how the R11, YES, Boudala et al. (2010), and UW algorithms differ from each other. As currently written, it is difficult for a reader unfamiliar with this sensor to understand the relationship between the R11 and YES algorithms. The manuscript is focused mainly on improving the R11 algorithm, but the connection between the R11 algorithm and the YES algorithm should be described more clearly. This will help establish the relevance of the manuscript to uninformed hotplate users, who will presumably rely upon the YES algorithm.

The algorithms presented in R11 and the present manuscript were adjusted for wind speed losses. There were also differences in the way the wind speed was estimated, and in how the conversion from power to latent energy ($f_2$) was estimated. With such possibly competing or self-compensating differences, a good reference precipitation measurement is necessary to properly evaluate the different algorithms. The present manuscript would be strengthened significantly if instead of adjusted single-Alter shielded weighing gauge precipitation measurements, reference precipitation measurements shielded by a double fence were used to validate the improved algorithm and compare it to the R11 and YES-derived precipitation. Particularly at windy sites, wind speed adjustments introduce significant uncertainty in the resultant precipitation measurements (Kochendorfer et al., 2017a;Fortin et al., 2008). The hotplate-derived wind speeds shown in the present manuscript were fairly low, and many of the E 98 values were greater than 0.5 (Table 5). This is good, but the manuscript should still include some discussion or quantification of the uncertainty introduced by the adjustment of the single-Altar shielded weighing gauge measurements. WMO-SPICE included three hotplates, tested for two winter seasons at

three separate sites with Double Fence Automated Reference (DFAR) measurements. I will try to help the authors obtain these data if they are interested in expanding the scope of the manuscript. I haven't looked at all of the WMO-SPICE hotplate data, but at the US (Marshall) site at least, the SHP values appear to be available. Feel free to contact me directly to discuss this at john.kochendorfer@noaa.gov.

5   Why were event totals used instead of higher frequency measurements? 30 or 60 minute rate/accumulation comparisons would provide many more points (or 'events') for evaluation, and would be accompanied with the added benefit of more stationary meteorological conditions and representative averages for wind speed, precipitation type, etc.

I didn't notice any mention of the hotplate power consumption. This should be added to the manuscript,
10   assuming that I did not overlook it. In the Introduction the advantages of the hotplate are carefully documented, but this significant limitation appears to be omitted.

How much testing of this sensor has been performed in rain? I would expect there to be a significant amount of splash out in heavy rain.

Ln 61. Did YES produce more than one type of hotplate during the history of this product? The version of the
15   hotplate firmware used should be included in the manuscript, if this relevant and available.

**Specific Comments:**

Ln 33 – 37. Heated tipping buckets are also used to measure snowfall (eg. Buisán et al., 2017).

Ln 56. Reference Fig. 1, which includes the radiation sensors. Also specify that only downwelling/incoming radiation was measured (and used to estimate net radiation). Or alternatively specify that the radiation sensors
20   only faced upward, and upwelling/outgoing radiation was not measured.

Ln 71. Here and elsewhere in the manuscript, the term "latent" should be replaced with "latent energy" or "latent heat". Likewise "sensible" (Ln 70) should be replaced with "sensible heat" throughout the manuscript.

Ln 72. State explicitly that the effects of radiation on the energy balance of the bottom plate are assumed to be negligible.

25   Ln 74. Change "evaluate at Reynolds number" to "estimate a Reynolds number".

Figure 1. I had a hard time identifying the air temperature sensor. I initially assumed that an independent measurement was used. This is in part because I am accustomed to seeing air temperature measurements within larger fan-aspirated or louvered radiation shields, but it would help if the labels on the right side of Fig. 1 were more clearly associated with their appropriate component. Also add RH and pressure to the appropriate
30   labels.

Ln. 97 – 98. Please summarize the formulation of the R11 conversion factors. Explain how they were different.

Ln. 110. Specify "downwelling longwave and shortwave fluxes".

Ln. 116 - 119. Radiative output and input haven't been defined. Also it needs to be made clear that radiative energy budget terms are only for the top plate. The sensible power output term is for both plates? $T_h$ is assumed equal for the top and bottom plate, even when precipitation is occurring? Has this been confirmed with actual measurements of the plate temperatures?

Ln. 133 – 134. "In the infrared… is the relevant property" is awkward as written.

Ln. 136. "The value we picked is…" is awkward as written.

Ln. 137. Specify that this is only for the top plate.

Ln. 141. "the hotplate's reflectance is the relevant property" is awkward as written.

Ln. 142. Specify that the shortwave flux was downwelling.

Ln. 164- 165. Add the word "accurate" and replace "whether the sensed hydrometeors are rain or snow" with, "precipitation type": Rewrite as, "rate is dependent on the *accurate* assessment of *precipitation type*."

Ln. 168. After "pressure" specify that these measurements were all recorded by the hotplate system.

Ln. 182. Please explain/explore why the calibration changed after servicing.

Ln. 188. Delete the word, "vertically" or rewrite. I understand what you are saying (after looking at Fig. 2), but is confusing because the plates are oriented horizontally.

Ln. 194  - 207. I worked it out eventually, but I found most of this section quite confusing. It should be made clear that this entire discussion is focused only on the top plate. It sounds like YES assumed that the source of upwelling longwave radiation (the ground) was the same temperature as the air. This is not a good assumption, as the surface temperature of the earth often differs significantly from the air. It would be worth comparing the resultant downwelling infrared radiation with a measurement recorded using a normal pyrgeometer, from which the incoming longwave flux is typically calculated using the radiation measurement and the body temperature of the sensor. It also begs the question of why YES took this extra step. Was it because they were interested in the net IR flux, or because the use the outgoing longwave flux in their assessment of the bottom plate energy balance? Also the "downward" in "net downward" (Ln. 194) should be deleted. Strictly speaking, "net downward" is an oxymoron. This may have contributed to my confusion. Because an upward facing pyrgeometer is not typically used to estimate net radiation without another downward facing pyrgeometer, I initially assumed that the "net" was incorrect, as opposed to the "downward".

Ln. 197. Add "net" – "Eq. 4 represents the *net* longwave radiant measurement…".

Ln. 207. Clarify that eq. 6 was only for use in the indoor experiment, where the temperature of the metal plate was estimated using the air temperature.

Ln. 212. Change the word, "settings" to something more appropriate like "variables".

5   Ln 214. Change the word, "with" to "using" or "from" – "$IR_d$ *was* calculated *using* Eq. 6".

Ln. 229. Rewrite as, "that relationship *was* applied…".

Ln. 230. Rewrite, "dimensionless representation of the sensible power output" - describe the Nusselt number more accurately.

Ln 251 – 252. Treating all precipitation above 0 °C as liquid is a little worrying. There are many examples of solid
10  precipitation occurring above 0 °C (and liquid precipitation occurring below 0 °C) (eg. Kochendorfer et al., 2017b;Wolff et al., 2015). A third conversion factor for mixed (or ambiguous) precipitation would be more defensible. It could be some combination of 9a and 9b, or a transition between the two.

Ln 268 and 271. Is it realistic to assume that the hotplate temperature ($T_h$) is equal to 0 °C? Is the bottom plate also 0 °C when precipitation is occurring? I thought that the temperature of both plates was nominally 75 °C.

15  Ln 282. "This is accounting" is awkward as written.

Ln 286. "catch efficiency is accounting" is awkward as written.

Ln. 287. "accounts for the fact" is awkward as written.

Ln. 300. I understand that this is being done for the sake of comparison with R11, but it should be pointed out that there is no good reason to adjust the hotplate derived wind speed to another height. The manuscript
20  should note that it is preferable to apply a catch efficiency function using the wind speed at the sensor's location.

Ln. 336. Why were only the hangar data used? The temperature range used in the different warm-cold tests varies significantly. In some cases it is quite narrow, and in others quite warm. How important or realistic is this?

25  Ln. 328 – 338, and Table 3. The derived hotplate temperature is quite variable. There appears to be some cross-correlation between gamma and the hotplate temperature (Table 3), with larger values of gamma associated with smaller hotplate temperatures. These values also appear to be correlated with the warm-cold temperatures of the indoor experiments they were derived from, which suggests that they may not be constant even for the same sensor. A comparison of measured hotplate temperatures (a small thermocouple or an IRT

could be used) and derived temperatures would help determine if the actual hotplate temperature varies as much as the derived temperature.

Ln. 361 – 362. This is awkward as written. Remove extraneous text. If there is no reason to question the fact that all of the water made it to the hotplate, there is no reason to bring it up.

5  Ln 371. UW is used to describe $P_{UW}$, the UW algorithm, and the UW hotplate. A different designation/abbreviation should be used to differentiate between sensor and algorithm. For example, $P_{UW}$ could easily be mistaken for $P$ from the UW hotplate, rather than $P$ from the UW algorithm. One solution would be to rename the UW hotplate.

Ln 374 and Fig. 4. R11 should be included in the drip test, and added to Fig. 4 and Fig. 5. Or the omission should
10  be justified in the text.

Ln 382 – 384 and Fig. 4. Augment the figure with cross-hatching or something similar to better illustrate the 1 min averaging periods. Also clarify that these 1-min periods were used for the regressions in Fig. 5, assuming that is what was done.

Fig. 5 and Ln 385 – 390. More detail is needed on how these values were obtained (see above). Also why were 0
15  mm hr$^{-1}$ precipitation periods excluded? An evaluation of the total accumulation should also be included. It is hard to tell from Fig. 4, but it seems possible that the 'overestimated' YES algorithm might be just as accurate as the UW algorithm after including the 0 precipitation periods and the period after the nondrip-to-drip transition. Also it isn't clear to me what role the minimum threshold plays here. In normal operations, I thought that a 0.2 mm hr$^{-1}$ threshold was used to differentiate between noise and precipitation, but Fig. 4 only seems to include a
20  0 mm hr$^{-1}$ threshold (to remove negative precipitation), and it is only for the YES sensor. Both the UW and the R11 algorithms include a threshold if I recall correctly. In normal operations how would the zero precipitation periods be handled for both algorithms? If I recall correctly, the YES sensors in SPICE had very few false-positives. The same methods recommended for normal field use should be employed in the evaluation, or at an explanation of why the thresholds weren't used should be added to the manuscript. The evaluation of the total
25  accumulation should be performed with the thresholds applied, although it could certainly also be performed without the thresholds to help demonstrate the potential effects of the threshold in normal field use.

Ln 404. Add an explanation of how events were defined. For example, more than x amount of precipitation, over x amount of time, beginning and ending with x minutes of zero precipitation... Also state whether the NOAH or the hotplate precipitation gauge was used to determine events.

30  Ln 414. Add "an" as follows: "and *an* upper-limit temperature…".

Ln 427 and 428. "$R_e$ extends smaller" and "there is an order of magnitude narrower $R_e$ range" are awkward as written.

Ln 441-442. Try to find a different term for "upwelling longwave". For many readers, the term "upwelling longwave radiation" already has a specific use that differs from the longwave radiation leaving the surface of the top hotplate. And throughout the manuscript the correct usage is "longwave radiation", not "longwave". The same rule applies to the use of "shortwave".

Ln 446. Rewrite as, "The first step in the calculation is *the* conversion of the latent *energy* term…".

Ln 447. Change "latent term" to "latent energy term" or "latent heat term".

Ln 448. One of the Methods Sections might be a more appropriate location than this Section, but a detailed explanation of this element-by-element vector multiplication should be added to the manuscript, including why it is necessary.

Ln 473 – 475. Explore the effects and uncertainty of the field-based calibration coefficients. How sensitive is precipitation to these? What happens if you swap them from site-to-site? Based only on their variability from site-to-site, there appears to be a significant amount of uncertainty in these terms. Calculate the effects of this uncertainty on precipitation.

Ln 483. "…ratios significantly smaller than unity" is awkward as written.

Ln 484. "…obtained when zeroing the shortwave term…" is awkward as written.

Ln 489. Add "the" to "values of *the* UW algorithm".

Ln 490. Change, "are detecting" to "detected".

Ln 498. Add "catch" to "event-averaged *catch* efficiency".

Ln 506. Change "*Es*" to "values of *E*" – *Es* could be mistaken as a separate term, rather than the plural form of *E*.

Ln 508. Delete "statistically" used at both the beginning and the end of this line.

Ln 521. Specify that the new radiation terms were only for the top plate.

Ln 524. Delete "have" in "we have used".

Ln. 565. Specify which component "Component of longwave flux" refers to. Based on Ln 118, it looks like it is the entire longwave flux, rather than a component.

Ln 596. Add *hp* (hotplate) to the list of subscripts.

Figure 3. Use consistent terminology. Change "New Algorithm" to "UW algorithm". Also in the caption explain that the Fig. 3b wind speeds were adjusted to account for the different heights.

**References:**

Buisán, S. T., Earle, M. E., Collado, J. L., Kochendorfer, J., Alastrué, J., Wolff, M., Smith, C. D., and López-Moreno, J. I.: Assessment of snowfall accumulation underestimation by tipping bucket gauges in the Spanish operational network, Atmos. Meas. Tech., 10, 1079-1091, 10.5194/amt-10-1079-2017, 2017.

Fortin, V., Therrien, C., and Anctil, F.: Correcting wind-induced bias in solid precipitation measurements in case of limited and uncertain data, Hydrological Processes, 22, 3393-3402, 10.1002/hyp.6959, 2008.

Kochendorfer, J., Nitu, R., Wolff, M., Mekis, E., Rasmussen, R., Baker, B., Earle, M. E., Reverdin, A., Wong, K., Smith, C. D., Yang, D., Roulet, Y. A., Meyers, T., Buisan, S., Isaksen, K., Brækkan, R., Landolt, S., and Jachcik, A.: Testing and development of transfer functions for weighing precipitation gauges in WMO-SPICE, Hydrol. Earth Syst. Sci. Discuss., 2017, 1-25, 10.5194/hess-2017-228, 2017a.

Kochendorfer, J., Rasmussen, R., Wolff, M., Baker, B., Hall, M. E., Meyers, T., Landolt, S., Jachcik, A., Isaksen, K., Brækkan, R., and Leeper, R.: The quantification and correction of wind-induced precipitation measurement errors, Hydrol. Earth Syst. Sci., 21, 1973-1989, 10.5194/hess-21-1973-2017, 2017b.

Wolff, M. A., Isaksen, K., Petersen-Overleir, A., Odemark, K., Reitan, T., and Braekkan, R.: Derivation of a new continuous adjustment function for correcting wind-induced loss of solid precipitation: results of a Norwegian field study, Hydrology and Earth System Sciences, 19, 951-967, 10.5194/hess-19-951-2015, 2015.

---

## Author Response (AR1)

- 1 This document contains:
- 2 1) a supplement to our online response to Professor Genthon (https://www.atmos-meas-tech-
- 3 discuss.net/amt-2017-234/amt-2017-234-AC2-supplement.pdf),
- 4 2) our response to Dr. Kochendorfer, and
- 5 3) the marked-up revised manuscript.

1) The introduction (lines 33 – 37) states that 2 types of instrumentations to measure precipitation have been developed: the capture and optical gauges. This 7 8 ignores radars which are powerful tools to measure and even profile precipitation. 9 Because of ground clutter and vertical resolution, radars do admittedly not measure precipitation right at the very surface but because they can profile vertically it may be 10 checked whether precipitation rates vary or not as it reaches closer to the surface. 11 12 Radars do not "obstruct the wind and deflects falling particles in the measurement zone" (line 38). There is no "clogging with snow" with radars (line 47). 13

In the US National Weather Service (NWS) network, a radar-derived snow amount is 14 15 dependent on gauge measurements made simultaneous with the radar measurements. A 16 published example of this is the NWS gauge-radar network data analyzed by Martinaitis et al. (2015). In that network, and presumably others, the gauge-calibrated radar-derived 17 estimates of snow can be significantly biased. Factors contributing to this are gauge 18 clogging by snowfall, postevent thaw of snow, and underestimation of snow because 19 20 wind speed is either not used to correct for gauge undercatch or it is unavailable. Martinaitis et al. (2015) state that "...The accuracy of hourly radar-derived QPE values of 21 winter precipitation is unknown..". Given this caveat, the dependence of radar 22 precipitation estimates on gauge measurements, and the factors you mention (vertical 23 structure, Earth curvature, ray ducting, ground clutter, and etc.), we have opted to not 24 25 mention that radars can be used to derive snowfall amounts.

The first sentence of the introduction was revised to this:

Two types of instrumentation are available for making point measurements of liquidequivalent snowfall rates and liquid-equivalent snow accumulations: 1) Weighing gauges and
related devices that measure snowfall as it collects in a container or on a surface (Brock and

Richardson, 2001; Chapter 9), and 2) optical gauges that measure the concentration and size
of snow particles either in free fall or within a wind tunnel (Loffler-Mang and Joss, 2000;

Deshler, 1988).

Martinaitis, S.M., S.B.Cocks, Y.Qi, B.T.Kaney, J.Zhang, and K.Howard, Understanding
winter precipitation impacts on automated gauge observations within a real-time
system, J. Hydrometeor., 16, 2345-2363, https://doi.org/10.1175/JHM-D-15-0020.1,

39

**40 We thank Dr. Kochendorfer for reviewing the manuscript.**

"Hotplate precipitation gauge calibrations and field measurements" describes 41 theoretical and observational work focused on a novel precipitation gauge. The hotplate 42 precipitation gauge measures precipitation by recording the amount of energy that is 43 consumed by heating, melting, and evaporating precipitation captured on an upward-facing 44 heated plate. The manuscript includes improvements to algorithms used to estimate the 45 wind-induced undercatch of the hotplate, the conversion of energy to evaporation, and the 46 47 effects of radiation, temperature, and wind on the energy balance and precipitation rate of 48 the sensor. These algorithms are developed and tested using field and laboratory 49 measurements.

Although the hotplate is not widely used, it is a unique, low-maintenance sensor capable of measuring all forms of precipitation in areas where power is available. Based on the hotplate's good performance in SPICE, there may be renewed interest in this technology when the SPICE results become widely available. The refinements and testing of the hotplate described in the present manuscript are therefore both valuable and timely. SPICE this is the World Meteorological Organization Solid Precipitation Intercomparison Experiment (e.g., Kochendorfer et al. 2017b).

Some general comments below indicate areas where there is room for improvement. 59 The manuscript is generally well written, but typos and other more specific suggestions are 60 documented in the specific comments.

**61 General Comments**

The description of the hotplates and algorithms discussed in the Introduction and 63 Methods sections should be augmented to clearly state how the R11, YES, Boudala et al. 64 (2010), and UW algorithms differ from each other. As currently written, it is difficult for a reader unfamiliar with this sensor to understand the relationship between the R11 and YES 65 66 algorithms. The manuscript is focused mainly on improving the R11 algorithm, but the 67 connection between the R11 algorithm and the YES algorithm should be described more 68 clearly. This will help establish the relevance of the manuscript to uninformed hotplate 69 users, who will presumably rely upon the YES algorithm.

On L268-L274, we assumed that the YES algorithm had incorporated R11's surface area
and R11's distinction between the theoretical and actual conversion factors. We don't
know for certain if these effects are incorporated into the YES algorithm. Until there is
more input on this, it will be impossible to establish the "connection" you are requesting.
Professor Rasmussen (https://www.atmos-meas-tech-discuss.net/amt-2017234/#discussion) speculated that YES had *not* incorporated the energy conversion factor
that R11 recommended, however, he was not categorical on this.

The algorithms presented in R11 and the present manuscript were adjusted for wind speed losses. There were also differences in the way the wind speed was estimated, and in how the conversion from power to latent energy (*f2*) was estimated. With such possibly competing or self-compensating differences, a good reference precipitation measurement
is necessary to properly evaluate the different algorithms.

The reviewer is neglecting the fact that we showed that a reference precipitation rate,
produced by the Ismatec pump, is consistent with rates derived using data from the two
hotplates, provided the data from those hotplates were processed using the UW algorithm.
This tests the *f*2 in a controlled setting.

The present manuscript would be strengthened significantly if instead of adjusted single-Alter shielded weighing gauge precipitation measurements, reference precipitation 87 88 measurements shielded by a double fence were used to validate the improved algorithm 89 and compare it to the R11 and YES-derived precipitation. Particularly at windy sites, wind 90 speed adjustments introduce significant uncertainty in the resultant precipitation 91 measurements (Kochendorfer et al., 2017a; Fortin et al., 2008). The hotplate-derived wind 92 speeds shown in the present manuscript were fairly low, and many of the E98 values were 93 greater than 0.5 (Table 5). This is good, but the manuscript should still include some 94 discussion or quantification of the uncertainty introduced by the adjustment of the single-95 Altar shielded weighing gauge measurements.

What we did discuss was this: In Fig. 12b there is agreement (statistically) when we compare

UW-algorithm-derived precipitation (based on U adjusted to 10 m for evaluating E) vs

NOAH-II precipitation. Also, in Fig. 12c there is agreement (statistically) when we compare

UW-algorithm-derived precipitation (based on U from the anemometer for evaluating E) vs

NOAH-II precipitation. In both of these comparisons the NOAH-II values are wind- corrected. We did not state that the agreement in Fig. 12b and Fig. 12c is relative to measurements that may be subject to bias (NOAH-II wind-corrected accumulations).

We will modify by finishing the paragraph with this:

Since there is error in the NOAH-II values used in this comparison, there is also need
for characterization of that uncertainty (random and systematic). Error can propagate from the
NOAH-II measurements themselves and from the catch efficiency function we applied to
those data (section 3.8).

WMO-SPICE included three hotplates, tested for two winter seasons at three 109 separate sites with Double Fence Automated Reference (DFAR) measurements. I will try to 110 help the authors obtain these data if they are interested in expanding the scope of the 111 manuscript. I haven't looked at all of the WMO-SPICE hotplate data, but at the US 112 (Marshall) site at least, the SHP values appear to be available. Feel free to contact me 113 directly to discuss this at john.kochendorfer@noaa.gov.

We feel this should be done, but not in this paper. Also, we thank you for your offer to share 115 data. If we do decide to do the analysis, we would want access to the hotplate gauge that was 116 used to acquire those measurements. This would enable us to do the hot/cold test, and thus 117 derive the surface temperature. The other ingredient of the analysis would be the calibration 118 of  $\alpha$ ,  $\beta$ , and  $\gamma$ . This requires a period of recorded SPICE data with varying wind, no 119 precipitation, and preferably at night.

Why were event totals used instead of higher frequency measurements? 30 or 60 minute rate/accumulation comparisons would provide many more points (or 'events') for evaluation, and would be accompanied with the added benefit of more stationary meteorological conditions and representative averages for wind speed, precipitation type, etc.

Our focus is on precipitation events lasting, on average, ~ 12 hours (Table 5). This provides
useful comparison of the two algorithms, and of the two gauges (hotplate and weighing).
Also, it is conjectural that precipitation is stationary on 30 minute time intervals. In our opinion, analysis of higher frequency measurements is beyond the scope of the manuscript;hence we plan to stick with our event-based comparison.

I didn't notice any mention of the hotplate power consumption. This should be
added to the manuscript, assuming that I did not overlook it. In the Introduction the
advantages of the hotplate are carefully documented, but this significant limitation appears
to be omitted.

Given that many weighing gauges have openings that are electrically heated, we are of the opinion that this is not a serious limitation for a hotplate. We added this to the end of L53:

In some applications a disadvantage of the hotplate, relative to a weighing gauge, isits electrical power consumption. This is ~ 200 W in Wyoming during winter.

How much testing of this sensor has been performed in rain? I would expect there tobe a significant amount of splash out in heavy rain.

We do not have any information on this. Rain rates in our four rain cases from OWL were ≤
mm/hr.

Ln 61. Did YES produce more than one type of hotplate during the history of this product? The version of the hotplate firmware used should be included in the manuscript, if this relevant and available.

L109. We revised the sentence to say this:

Consequently, our hotplate (Wolfe and Snider, 2012) was upgraded to firmware147 version 3.1.2 in 2011.

Specific Comments:

Ln 33 – 37. Heated tipping buckets are also used to measure snowfall (eg. Buisán et
al., 2017).

L34-35. Because the following reference provides a general overview of many differentdevices, including tipping buckets, we revised the sentence to say this:

- 153 1) Weighing gauges and related devices that measure snowfall as it collects in a 154 container or on a surface (Brock and Richardson, 2001; Chapter 9),
- Brock, F.V., and S.J. Richardson, Meteorological Measurement Systems, Oxford
  University Press, New York, 304 pp., 2001

Ln 56. Reference Fig. 1, which includes the radiation sensors. Also specify that only downwelling/incoming radiation was measured (and used to estimate net radiation). Or alternatively specify that the radiation sensors only faced upward, and upwelling/outgoing

- 160 radiation was not measured.
- 161 This was remedied in the following ways:
- 162 Changed L142 to "a measured downwelling shortwave flux (SW; Table 1)"
- 163 Changed entry in Table 1 to "Downwelling Shortwave Flux"
- 164 Changed entry in Table 1 to "Longwave Radiation"
- 165 Removed two occurrences of "net" in the text

Ln 71. Here and elsewhere in the manuscript, the term "latent" should be replaced with "latent energy" or "latent heat". Likewise "sensible" (Ln 70) should be replaced with "sensible heat" throughout the manuscript. 169 We reserved "latent heat" for the quantity of energy absorbed during a phase transition. This is consistent with the definition in the American Meteorological Society Glossary: "The 170 specific enthalpy difference between two phases of a substance at the same temperature." 171 In the revision, L71 and L215, and throughout, we changed "latent energy" to "latent power 172 output." It's dorky, but necessary because the budget equation is an energy-rate equation. 173 174 Electrical power supplied to the top plate  $(Q_{top})$  compensates for power lost via sensible energy, radiative, and vapor mass transfer. Henceforth, we refer to the latter process 175 176 as latent power output. In Eq. 3 we have "power", not "energy", so in L440 to L444 we modified the text to this: 177 Fig. 8 shows budget terms (Eq. 3) for one of the four rainfall events in our dataset 178 179 (OWL-15). The three power output terms (sensible, longwave, and latent), and three power input terms (top plate, longwave, and shortwave) are shown in Fig. 8a - b. In this section we 180 181 begin with the sequence of latent power output ( $P \cdot E/f_2$  in Fig. 8a) and explain how we calculate the sequence of rainfall rate. 182

And the caption to Fig. 8 was changed to this:

Figure 8 – Hotplate properties during rain (event = OWL-15). Because this event
classifies as rain, *E* = 1 was applied in the UW algorithm. a) Power output terms in the Eq. 3;
i.e., the sensible, latent, and longwave output terms. b) Power input terms in the Eq. 3; i.e.,
the top plate, longwave, and shortwave input terms. The shortwave term is zero for this
nighttime example, but is set to 0.1 W in the plot. c) Thresholded precipitation rate. d)
Unthresholded precipitation rate.

Ln 72. State explicitly that the effects of radiation on the energy balance of thebottom plate are assumed to be negligible.

**192 L71 to L73. We revised this text to say this:**

The hotplate-derived wind speed, evaluated at gauge height via the "factory" 193 calibration" discussed in R11, is used in this analysis. The bottom plate power  $(Q_{bot})$  is likely 194 a measurement used in the calculation of that wind speed, but this is speculative because the 195 factory wind speed algorithm is proprietary. We symbolize this wind speed as U and use it to 196 197 evaluate a Reynolds number (*Re*), and use the latter to parameterize sensible heat transfer 198 from the ventilated surface of the top plate. R11 also derived wind speeds by fitting  $Q_{bot}$ , 199 ambient temperature, and a wind speed measured at 10 m above ground level (AGL). This 200 wind speed is not used in this analysis. The hotplate ambient temperature (T) measurement 201 comes from the sensor seen below the radiation instruments (Fig. 1), the relative humidity 202 (*RH*) measurement comes from a sensor that protrudes below the electronics box (Fig. 1), and 203 the hotplate pressure sensor is contained within the electronics box. A complete description of our nomenclature is provided in the Appendix. 204

Ln 74. Change "evaluate at Reynolds number" to "estimate a Reynolds number".

Are you commenting on this because we held properties of the film constant in the calculation of the Reynolds and Nusselt numbers (section 3.6), in the fitting (section 5.2), and in the application of the fitted relationship within the UW algorithm (section 5.3)? In our opinion, constant film properties is an appropriate assumption provided the assumption made in the fitting and is also made in the application of the fit. No change was made.

Figure 1. I had a hard time identifying the air temperature sensor. I initially assumed that an independent measurement was used. This is in part because I am accustomed to seeing air temperature measurements within larger fan-aspirated or louvered radiation shields, but it would help if the labels on the right side of Fig. 1 were more clearly associated with their appropriate component. Also add RH and pressure to the appropriatelabels.

Please see above, and also, see that we added this section to the Methods:

**218** 3.1 - Temperature Measurements

Ice bulb temperatures at OWL were calculated using temperature, RH, and pressure 219 measurements made within a fully shielded housing (Steenburgh et al., 2014). At GLE and 220 221 BTL ice bulb temperatures were calculated using the hotplate-derived temperature, RH, and pressure values (Table 1). Because the hotplate temperature sensor is incompletely shielded 222 223 (Fig. 1), there is concern that its measurement is positively biased by solar heating. We investigated this by differencing hotplate-derived temperatures, acquired during precipitation 224 225 events at OWL, and values acquired by the fully shielded temperature sensor operated at 226 OWL. On average, the hotplate values were larger  $(0.4 \pm 0.4 \text{ °C})$ . We did not attempt to 227 correct for this bias.

- Ln. 97 98. Please summarize the formulation of the R11 conversion factors. Explain
   how they were different.
- 230 The requested information is presented in a subsequent section.

Ln. 110. Specify "downwelling longwave and shortwave fluxes".

Yes.

Ln. 116 - 119. Radiative output and input haven't been defined. Also it needs to be 234 made clear that radiative energy budget terms are only for the top plate.

The sentence introducing Eq. 3 is changed to this:

We used the following equation to analyze the top plate's power budget.

| 237 | Also, in our o | pinion, 1 | L117, | L118, and | d L119, | are self-exp | planatory. |
|-----|----------------|-----------|-------|-----------|---------|--------------|------------|
|-----|----------------|-----------|-------|-----------|---------|--------------|------------|

- 238 The sensible power output term is for both plates?
- 239 This is clarified in the revision.
- 240 *Th* is assumed equal for the top and bottom plate, even when precipitation is occurring?

We evaluated the surface temperature of the top plate in the "Warm-cold Ambient

Temperature Tests" section. We did not model the *bottom* plate power budget, or derive the bottom plate temperature. We presume the bottom plate temperature is a parameter in the

"factory calibration" discussed previously. However, the factory calibration is proprietary, so

- this is just speculation.
- 247 Has this been confirmed with actual measurements of the plate temperatures?

Not that we are aware of. Please see previous response.

Ln. 133 – 134. "In the infrared... is the relevant property" is awkward as written.

Revised.

Ln. 136. "The value we picked is..." is awkward as written.

Revised.

Ln. 137. Specify that this is only for the top plate.

We changed the sentence on L133 to read as this:

Two radiative properties are applied in our analysis of the top plate's power budget (Eq. 3). In the infrared, or longwave, the emissivity of the top plate is the key property.

Ln. 141. "the hotplate's reflectance is the relevant property" is awkward as written. 257 Revised. 258 Ln. 142. Specify that the shortwave flux was downwelling. 259 Yes. 260 Ln. 164-165. Add the word "accurate" and replace "whether the sensed 261 hydrometeors are rain or snow" with, "precipitation type": Rewrite as, "rate is dependent 262 on the *accurate* assessment of *precipitation type*." 263 Thanks for the suggestion. In our view, "precipitation type" is too vague. The text was 264 changed to this: 265 The accuracy of a hotplate-estimated precipitation rate depends on whether the sensed 266 hydrometeors are rain or snow (R11 and Fig. 3a). 267 Ln. 168. After "pressure" specify that these measurements were all recorded by the 268 269 hotplate system. Since this is explained in the added section 3.1, the sentence was changed to this: 270 271 Measurements used to derive the ice bulb temperature are described in section 3.1. 272 Ln. 182. Please explain/explore why the calibration changed after servicing. 273 YES told us that electrical components were replaced, however, we feel that this is too much detail to present. Adam Wettlaufer did state that calibration changes "...were likely a 274 consequence of the servicing provided by YES.." Interested readers can look at his analysis. 275 In the revision we will revise this sentence to this: 276 277 Wettlaufer (2013) demonstrates that calibration constants did change over the 2011 to

2015 interval, and likely in response to servicing conducted twice at YES.

Ln. 188. Delete the word, "vertically" or rewrite. I understand what you are saying 280 (after looking at Fig. 2), but is confusing because the plates are oriented horizontally.

Yes.

Ln. 194 - 207. I worked it out eventually, but I found most of this section quite confusing. It should be made clear that this entire discussion is focused only on the top plate.

In the revision, we have made it clear that Eq.1 and Eq. 3 are power budgets for the top plate.

It sounds like YES assumed that the source of upwelling longwave radiation (the
ground) was the same temperature as the air. This is not a good assumption, as the surface
temperature of the earth often differs significantly from the air.

A fair criticism, but we compensate for this in Eq. 5. Also, the L193 to L195 sentence wasrevised to this:

In response to that finding, newer versions of the hotplate have a device that measures
longwave radiation (pyrgeometer, e.g., Albrecht et al., 1974) (Fig. 1 and Table 1).

It would be worth comparing the resultant downwelling infrared radiation with a 294 measurement recorded using a normal pyrgeometer, from which the incoming longwave 295 flux is typically calculated using the radiation measurement and the body temperature of 296 the sensor. It also begs the question of why YES took this extra step. Was it because they 297 were interested in the net IR flux, or because the use the outgoing longwave flux in their 298 assessment of the bottom plate energy balance? Also the "downward" in "net downward" 299 (Ln. 194) should be deleted. Strictly speaking, "net downward" is an oxymoron. This may 300 have contributed to my confusion. Because an upward facing pyrgeometer is not typically used to estimate net radiation without another downward facing pyrgeometer. I initially 301 302 assumed that the "net" was incorrect, as opposed to the "downward". Sorry for the confusion. The word "net" is removed from the revision. 303 Ln. 197. Add "net" – "Eq. 4 represents the *net* longwave radiant measurement...". 304 We are not interested in the "net" longwave. So, rather than explain Eq. 4 as a "net", we 305 306 explained the two quantities that contribute to  $M_{\rm IR}$ . Also, please see above; the word "net" is removed from the revised manuscript. 307 308 Ln. 207. Clarify that eq. 6 was only for use in the indoor experiment, where the temperature of the metal plate was estimated using the air temperature. 309 This is explained later in the text; in our opinion this is not the place for that explanation. 310 Ln. 212. Change the word, "settings" to something more appropriate like "variables". 311 In the revision, we use "calibration parameters." Later in the discussion, these become a 312 " $T_h/\gamma$  pair." 313 Ln 214. Change the word, "with" to "using" or "from" – "IRd was calculated using Eq. 314 6". 315 316 Yes. Ln. 229. Rewrite as, "that relationship was applied...". 317 318 Yes. Ln. 230. Rewrite, "dimensionless representation of the sensible power output" -319

describe the Nusselt number more accurately.

**321 This was revised for clarity.**

Ln 251 – 252. Treating all precipitation above 0 °C as liquid is a little worrying. There are many examples of solid precipitation occurring above 0 °C (and liquid precipitation occurring below 0 °C) (eg. Kochendorfer et al., 2017b; Wolff et al., 2015). A third conversion factor for mixed (or ambiguous) precipitation would be more defensible. It could be some combination of 9a and 9b, or a transition between the two.

In our opinion, the cases we analyze are clear cut. That is, four have ice bulb temperatures larger than 0 °C, and thus the particles are either melted or melting as they approach the hotplate surface. Hence, for our cases we do not think we need to justify the "combination"

you are recommending.

Ln 268 and 271. Is it realistic to assume that the hotplate temperature (*Th*) is equal to 0 °C?

We **assumed** Th =  $0 \,^{\circ}$ C (see L255) to be consistent with R11. We did this so we could compare to the f2 reported in R11. We concluded that our calculation of f2, and R11's calculation of f2, are consistent. In both cases there is the assumption  $T_h = 0$  °C; however, we feel the proper way to do this is to account the warming terms involving heat capacities.

These issues are discussed in section 3.8 (revision), and in section 3.7 (reviewed manuscript).

Is the bottom plate also 0 °C when precipitation is occurring?

We did not model the *bottom* plate power budget, or derive the bottom plate temperature.

I thought that the temperature of both plates was nominally 75 °C.

This is a simplification introduced by R11. But it is restricted to their formulation of f2, which we mimicked (see above).

- Ln 282. "This is accounting" is awkward as written.
- 344 We revised the sentence to this:
- This is due to the warming discussed in section 2.
- Ln 286. "catch efficiency is accounting" is awkward as written. Ln. 287. "accounts for
- 347 the fact" is awkward as written.
- 348 The two sentences were rewritten to this:

In this section, we evaluate a wind speed-dependent function and use it to account for the top plate's snow particle catch efficiency (*E*; section 2). The physical processes this function accounts for are, 1) snow particle bouncing subsequent to collision with the top plate, followed by transfer away from the top plate by wind, and 2) shearing off of a snow particle after it has landed on the top plate (R11).

Ln. 300. I understand that this is being done for the sake of comparison with R11, but it should be pointed out that there is no good reason to adjust the hotplate derived wind speed to another height. The manuscript should note that it is preferable to apply a catch efficiency function using the wind speed at the sensor's location.

- 358 This is later in manuscript. Please see L514.
- Ln. 336. Why were only the hangar data used?
- 360 We changed this text to the following:

In our analysis of the warm-cold measurements we only used data acquired in the 362 hangar. As we describe below, this may have improved the accuracy of the resultant  $T_h/\gamma$ 363 pairs. This is because all data needed to derive a  $T_h/\gamma$  pair can be obtained without turning off 364 the hotplate. Wettlaufer (2013) analyzed both hanger and lab data. Both in his work and in

- ours, the relevant hotplate properties were derived by averaging over a 5 minute warm interval and a 5 minute cold interval, and applying these averages in Eq. 7a - 7b.
- 367 The temperature range used in the different warm-cold tests varies significantly. In 368 some cases it is quite narrow, and in others quite warm.
- We did not have control over how cold the hanger gets, at night, or how warm it gets, during the day. The warm-cold temperature pairings (hangar) came from measurements made in the middle of the afternoon and early in the morning, respectively.
- Ln. 328 338, and Table 3. The derived hotplate temperature is quite variable.
- We did discuss the servicing done at YES (L178 to L184). This may be the source of the variability. Also, some of the variability may stem from the  $\pm 0.5$  W error. The latter is the basis for the error limits we placed on the Th and  $\gamma$  values presented in Table 3. For these error limits, please see Table 3 and the discussion on L349 to L353.
- There appears to be some cross- correlation between gamma and the hotplate 377 378 temperature (Table 3), with larger values of gamma associated with smaller hotplate temperatures. These values also appear to be correlated with the warm-cold temperatures 379 380 of the indoor experiments they were derived from, which suggests that they may not be 381 constant even for the same sensor. A comparison of measured hotplate temperatures (a 382 small thermocouple or an IR could be used) and derived temperatures would help 383 determine if the actual hotplate temperature varies as much as the derived temperature. Note that the second term in Eq. 7a or Eq. 7b scales with the product of  $T_h$  and  $\gamma$ ; hence, the 384 385 correlation you mention is a consequence of the mathematical form of the Equations 7a and
- 386 7b. We do not think it is important to go into that issue.

We have not attempted to measure the surface temperature using the methods yourecommend.

Ln. 361 – 362. This is awkward as written. Remove extraneous text. If there is no reason to question the fact that all of the water made it to the hotplate, there is no reason to bring it up.

Yes.

Ln 371. UW is used to describe *PUW*, the UW algorithm, and the UW hotplate. A different designation/abbreviation should be used to differentiate between sensor and algorithm. For example, *PUW* could easily be mistaken for *P* from the UW hotplate, rather than *P* from the UW algorithm. One solution would be to rename the UW hotplate. In the revision, we clarify this earlier in the manuscript. This revised text and footnote are near L58: These are a hotplate gauge owned by the University of Wyoming (UW) and by the

These are a hotplate gauge owned by the University of Wyoming (UW) and by theNational Center for Atmospheric Research (NCAR; Boulder, CO).

When a distinction is needed, we indicate the hotplate, followed by a forward slash,
and the location of the deployment. For example, the UW hotplate, deployed at the OWL site,
is designated UW/OWL.

Ln 374 and Fig. 4. R11 should be included in the drip test, and added to Fig. 4 and Fig.
5. Or the omission should be justified in the text.

First, the setting E = 1 is applied here. That is definitely true for the PUW and presumably true 407 for the PYES. Hence, a difference cannot arise from different catch efficiencies. Second, the  $f_2$

we apply in the new algorithm is the solid line in Fig. 3a  $(2.92 \times 10^{-8} \text{ m/J})$ , and we are assuming that the  $f_2$  applied in the YES algorithm is the dashed line (3.38x10--8 m/J). We 410 evaluated both of these f2 values at the right-hand margin of Fig. 3a (T = 5 °C). Based on the 411  $f_2$  discussed in the previous two sentences, we expect the PREF/PYES ratio to be ~ 16% smaller 412 than the PREF/PUW ratio. This is what we see in Fig. 5, so we conclude as described on L392 413 to L399. Adding what you are proposing will not change these conclusions, but it will 414 complicate interpretation of Fig. 4 and Fig. 5. We prefer to avoid inserting information you 415 are requesting, both in the figure and in the discussion of the figure.

Ln 382 – 384 and Fig. 4. Augment the figure with cross-hatching or something similar
to better illustrate the 1 min averaging periods.

The abscissa is ticked in minutes. Perhaps that was missed. Adding hatching, or similar,complicates the figure.

We will revise the caption of Fig. 5 to this:

Figure 4 – Precipitation rates, derived using the UW and YES algorithms, plotted vs time. Dashed vertical lines illustrate nondrip-to-drip transitions, drip-to-nondrip transitions, and one-minute precipitation averaging intervals. In this figure, the one-minute averaging intervals are ~ 16:08 to ~ 16:09 UTC and ~ 16:17 to ~ 16:18 UTC. Measurements are from the UW hotplate operating indoors on 20120229. The UW/GLE calibration constants (Table 3) and an  $f_2$  derived with the second of two methods (section 3.7) were applied in the UW algorithm.

Also clarify that these 1-min periods were used for the regressions in Fig. 5, assuming 429 that is what was done.

Yes, here is how we changed L382 to L383:

We set the end of these at the drip-to-nondrip transitions and symbolize the averages as  $\langle P_{UW} \rangle$  and  $\langle P_{YES} \rangle$ .

Fig. 5 and Ln 385 – 390. More detail is needed on how these values were obtained
(see above).

This is addressed (see above).

Also why were 0 mm hr-1 precipitation periods excluded? An evaluation of the total accumulation should also be included. It is hard to tell from Fig. 4, but it seems possible that the 'overestimated' YES algorithm might be just as accurate as the UW algorithm after including the 0 precipitation periods and the period after the nondrip-to-drip transition.

The Fig. 4 shows, based on one-minute averaging, that YES is larger than UW. If an integral from the nondrip-to-drip transition to the drip-to-nondrip transition is performed, it seems, from Fig. 4, that the YES/UW difference will enhance further. Yet, because of the possibility of a violation of the steady-state assumption (L376), and because of the minimum (L377), we discourage such a YES/UW comparison.

Careful here with "accuracy." We have a reference, Pref. We use the latter for the determination of accurate vs inaccurate. Fig. 5 shows inaccuracy for YES, and accuracy for

UW. Table 4 summarizes the accuracy/inaccuracy of two hotplate gauges.

About the "zeros." The graphs on the next page have the result presented in Fig. 5, but with 449 UW algorithm in top left, the YES algorithm in the bottom left, the UW algorithm with zeros 450 included (top right), and the YES algorithm with zeros included (bottom right). Clearly, the 451 effect you are speculating about ("…seems possible…") is negligible. Finally, because we 452 take  $P_{ref}$  to be a standard, the curve fitting is based on minimization of the sum of the squares 453 of the *x* deviations (horizontal departures of data from the regression line). This is stated in 454 the Fig. 5 caption.

**C:\jeff\thesis\_adam\dat2\GLE\_drip\drip\_GLE\_unventilated.txt**